# TrpV1 receptor activation rescues neuronal function and network gamma oscillations from Aβ-induced impairment in mouse hippocampus in vitro

Hugo Balleza-Tapia[1], Sophie Crux[1,2], Yuniesky Andrade-Talavera[1], Pablo Dolz-Gaiton[1], Daniela Papadia[1], Gefei Chen[3], Jan Johansson[3], André Fisahn[1]*

[1]Neuronal Oscillations Laboratory, Department of Neurobiology, Care Sciences and Society, Center for Alzheimer Research, Neurogeriatrics Division, Karolinska Institutet, Stockholm, Sweden; [2]German Center for Neurodegenerative Diseases, Munich, Germany; [3]Department of Neurobiology, Care Sciences and Society, Center for Alzheimer Research, Neurogeriatrics Division, Karolinska Institutet, Stockholm, Sweden

**Abstract** Amyloid-β peptide (Aβ) forms plaques in Alzheimer's disease (AD) and is responsible for early cognitive deficits in AD patients. Advancing cognitive decline is accompanied by progressive impairment of cognition-relevant EEG patterns such as gamma oscillations. The endocannabinoid anandamide, a TrpV1-receptor agonist, reverses hippocampal damage and memory impairment in rodents and protects neurons from Aβ-induced cytotoxic effects. Here, we investigate a restorative role of TrpV1-receptor activation against Aβ-induced degradation of hippocampal neuron function and gamma oscillations. We found that the TrpV1-receptor agonist capsaicin rescues Aβ-induced degradation of hippocampal gamma oscillations by reversing both the desynchronization of AP firing in CA3 pyramidal cells and the shift in excitatory/inhibitory current balance. This rescue effect is TrpV1-receptor-dependent since it was absent in TrpV1 knockout mice or in the presence of the TrpV1-receptor antagonist capsazepine. Our findings provide novel insight into the network mechanisms underlying cognitive decline in AD and suggest TrpV1 activation as a novel therapeutic target.

DOI: https://doi.org/10.7554/eLife.37703.001

*For correspondence:
andre.fisahn@ki.se

**Competing interests:** The authors declare that no competing interests exist.

## Introduction

Alzheimer's disease (AD) is an irreversible progressive neurodegenerative disorder and the most common type of dementia in the elderly (*Brookmeyer et al., 2007*; *Qiu et al., 2009*). AD is characterized by a progressive deficit in cognition, principally in memory and learning and has become a major public health challenge. Today, there are several lines of evidence indicating that amyloid-β peptide (Aβ), the main component of senile plaques in AD, plays an important role in AD pathogenesis and is responsible for the early cognitive deficit observed in the disease with the Aβ1–42 peptide thought to be particularly disease relevant (*Lue et al., 1999*; *Näslund et al., 2000*; *Walsh and Selkoe, 2007*). Aβ has also been shown to induce impairments in neuronal network functions that are important for cognition, such as network oscillations (*Driver et al., 2007*; *Nerelius et al., 2009*; *Balleza-Tapia et al., 2010*; *Villette et al., 2010*; *Gutiérrez-Lerma et al., 2013*; *Kurudenkandy et al., 2014*).

Neuronal network oscillations in various frequency bands are crucial for information processing in the brain (*Buzsáki and Draguhn, 2004*). In particular, gamma-frequency oscillations (30–80 Hz) have been suggested to underlie cognitive processes such as attention, sensory perception and memory (*Tallon-Baudry and Bertrand, 1999*; *Sederberg et al., 2007*; *Fries, 2015*). This is supported by the finding that gamma oscillations progressively decrease in AD patients in lock-step with advancing cognitive impairment (*Herrmann and Demiralp, 2005*; *Koenig et al., 2005*; *de Haan et al., 2012*).

Endocannabinoids such as anandamide have been shown to possess protective properties against Aβ-induced neuronal damage (*Aso and Ferrer, 2014*). Indeed, a negative correlation was found between anandamide and Aβ levels in AD brains (*Jung et al., 2012*) suggesting that the endocannabinoid's protective role is decreased in AD. In this context, it has been reported that enhancement of anandamide levels by inhibition of endocannabinoid cellular reuptake can reverse hippocampal damage and memory impairment in rodents pretreated with Aβ (*van der Stelt et al., 2006*). Also, in vitro studies have shown that anandamide can protect neurons from Aβ's neurotoxic effects (*Milton, 2002*) through a mechanism independent of CB1 and CB2 activation (*Harvey et al., 2012*). Since anandamide also activates the transient receptor potential cation channel/vanilloid receptor TrpV1 (*Aso and Ferrer, 2014*), it is possible that its protective properties are mediated by this receptor.

TrpV1 is a ligand-gated non-selective cation channel and belongs to the transient receptor potential (TRP) channel family (*Nieto-Posadas et al., 2011*). It is known to be activated by capsaicin (Cp; *Caterina et al., 1997*), the pungent ingredient in hot chili peppers, but also by numerous stimuli including several endogenous lipid molecules (endovanilloids) such as anandamide. TrpV1 has been reported to be expressed in the brain (*Sasamura et al., 1998*; *Mezey et al., 2000*; *Roberts et al., 2004*; *Tóth et al., 2005*; *Cristino et al., 2006*) where it is involved in several functions such as the modulation of spine morphology, synaptic transmission and plasticity (*Ho et al., 2012*; *Edwards, 2014*; *Martins et al., 2014*).

Previously, we have established that Aβ-induced degradation of gamma oscillations in the hippocampal network is caused by the desynchronization of action potential (AP) firing of pyramidal cells (PC) and a shift in the excitatory/inhibitory current balance (*Kurudenkandy et al., 2014*). Since anandamide has been reported to have protective properties against Aβ-induced impairments in vitro and in vivo both in WT and AD animal models (*Aso and Ferrer, 2014*), we wanted to investigate the feasibility of cannabinoid and/or TrpV1 receptor activation as a therapeutic avenue for prevention of, and more importantly rescue from, Aβ-induced degradation of cellular mechanisms and functional network dynamics important for cognition.

## Results

### Cannabinoid receptor activation fails to prevent Aβ-induced impairment of functional network dynamics

Previously, we reported that a physiologically relevant concentration of Aβ1–42 (Aβ hereafter; *Roher et al., 2009*) reduces gamma oscillations in the hippocampal network by a desynchronization of PC AP and by a shift in the excitatory/inhibitory current balance (*Kurudenkandy et al., 2014*). Since endocannabinoids such as anandamide have been reported to have preventive properties against Aβ-induced impairments in WT and AD animal models in vitro and in vivo (*Aso and Ferrer, 2014*), we first investigated whether cannabinoid receptor (CB1 and CB2) activation could prevent the Aβ-induced reduction of gamma oscillations.

Hippocampal slices were incubated for 15 min with 50 nM Aβ and 200 nM of the CB1/CB2 agonist WIN. After incubation, gamma oscillations in area CA3 were induced by bath perfusion of 100 nM KA (20 min) in an interface recording chamber. We found that compared to hippocampal slices in control conditions (100 nM KA, power: $11.6 \pm 1.7 \times 10^{-9}$ V$^2$, n = 32, *Figure 1A–B*), Aβ+WIN incubation resulted in significantly reduced gamma oscillation power ($64.2 \pm 9.0\%$ reduction, power: $4.1 \pm 1.0 \times 10^{-9}$ V$^2$, n = 18, p = 0.0002 vs. control, *Figure 1A–B*), that was not significantly different from the reduced gamma power recorded in slices incubated with Aβ only ($59.4 \pm 8.9\%$ reduction, power: $4.7 \pm 1.0 \times 10^{-9}$ V$^2$, n = 20, p = 0.0007 vs control, *Figure 1A–B*).

Likewise, when incubating hippocampal slices with 50 nM Aβ and a specific agonist for CB1 (ACEA, 200 nM, 15 min), we found that gamma oscillations continued to show a significant reduction

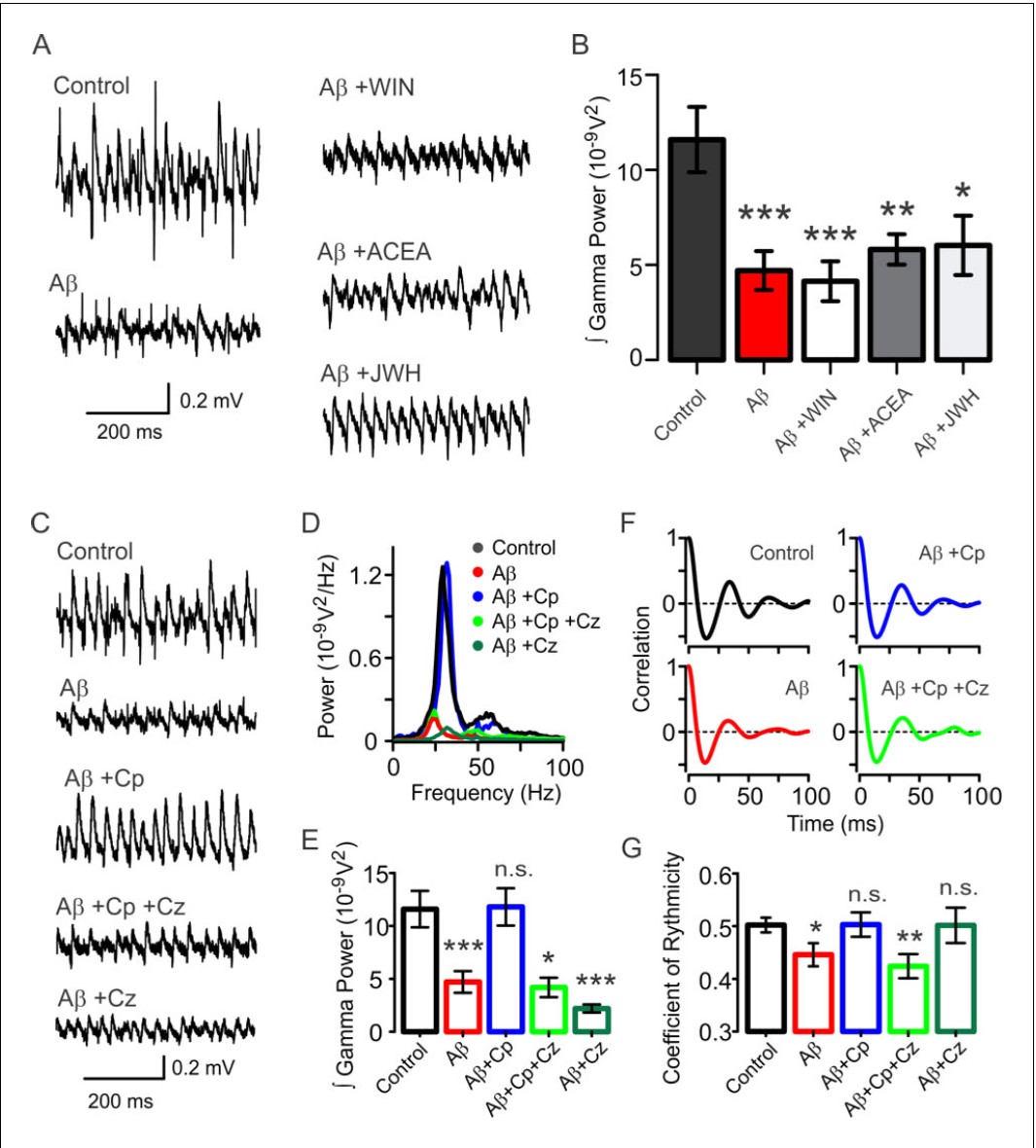

**Figure 1.** Activation of TrpV1, but not CB1/CB2, protects the hippocampal network from Aβ-induced degradation of gamma oscillations. (**A**) Representative sample traces of KA-induced gamma oscillations in hippocampal slices. Control: gamma oscillations 20 min after 100 nM KA bath application; Aβ: degradation of KA-induced gamma oscillations after 15 min pre-incubation with 50 nM Aβ; Aβ+WIN: pre-incubation with CB1/CB2 receptor agonist WIN does not protect gamma oscillations from Aβ-induced degradation; Aβ+ACEA: CB1 agonist ACEA; Aβ +JWH: CB2 agonist JWH; both also show no protective properties against the Aβ-induced degradation. (**B**) Summary histogram of integrated gamma power from the experimental conditions described in A. (**C**) Representative sample traces of control gamma oscillation (control) and the Aβ-induced reduction of gamma oscillations (Aβ). TrpV1 activation by its agonist Cp (10 μM) protected gamma oscillations from Aβ-induced impairment (Aβ+Cp) while TrpV1 antagonist Cz (10 μM) blocked Cp protection (Aβ+Cp+Cz). The effect of Aβ together with Cz is also shown (Aβ+Cz). (**D**) Power spectra for the experimental conditions described in C: Control (black), Aβ (red), Aβ+Cp (blue), Aβ+Cp+Cz; (green), Aβ+Cz; (dark green). (**E**) Integrated power of gamma oscillations from the experiments described in C and D, showing the protective properties of TrpV1 activation by Cp. (**F**) Average auto-correlograms showing the Aβ-induced reduction of the quality of gamma oscillation and the protective effect following TrpV1 activation. (**G**) Summary bar-graph of the coefficient of rhythmicity (Cr) calculated from the experimental conditions described in F. The effect of Aβ together with Cz is also shown (Aβ+Cz). Quantification of integrated power and Cr were performed from the same 60s-segment 20 min after gamma oscillations induction by bath application of 100 nM KA (for experimental data see *Figure 1—source data 1*).

*Figure 1 continued on next page*

*Figure 1 continued*

Mann-Whitney test (one-tailed) was used for statistical significance on absolute values. Data is presented as mean ± SEM. * indicates p < 0.05, **p < 0.01, ***p < 0.001 and n.s. indicates no statistically significant differences.
DOI: https://doi.org/10.7554/eLife.37703.002

The following source data and figure supplements are available for figure 1:

**Source data 1.** Experimental data for *Figure 1*.
DOI: https://doi.org/10.7554/eLife.37703.005

**Figure supplement 1.** Capsaicin induces a reduction in hippocampal gamma oscillation power via a TrpV1-independent mechanism.
DOI: https://doi.org/10.7554/eLife.37703.003

**Figure supplement 1—source data 1.** Experimental data for *Figure 1—figure supplement 1*.
DOI: https://doi.org/10.7554/eLife.37703.004

in power (49.9 ± 6.9% reduction, power: 5.8 ± 0.8×$10^{-9}$ V$^2$, n = 21, p = 0.0062 vs. control, *Figure 1A–B*). Similar results were obtained when a specific CB2 agonist (JWH; 100 nM) was used (48.0 ± 13.5% reduction, power: 6.0 ± 1.6×$10^{-9}$ V$^2$, n = 10, p = 0.0265 vs control, *Figure 1A–B*). Even when increasing WIN concentration to 1 µM no protective effect against Aβ-induced gamma oscillation reduction was observed (58.9 ± 15.4% reduction, power: 4.8 ± 1.8×$10^{-9}$ V$^2$, n = 5, p = 0.0326 vs control). All together, these data indicate that cannabinoid receptor activation does not exert a protective effect against Aβ-induced impairment of functional network dynamics in hippocampus.

## TrpV1 receptor activation prevents Aβ-induced impairment of functional network dynamics

*Harvey et al. (2012)* have reported that the protective effect of the endocannabinoid anandamide against Aβ is independent of CB1 and CB2 receptors activation. Since anandamide is also an agonist of the TrpV1 receptor (*Aso and Ferrer, 2014*), we proceeded to investigate whether selective TrpV1 activation would result in protection against Aβ-induced impairment of functional network dynamics. In contrast to cannabinoid receptor activation, we found that incubation with 50 nM Aβ and 10 µM of the TrpV1 receptor agonist capsaicin (Cp, 15 min) completely prevented the Aβ-induced reduction of gamma power (control: 11.6 ± 1.7×$10^{-9}$ V$^2$, n = 32; Aβ+Cp: 11.8 ± 1.8×$10^{-9}$ V$^2$, n = 18; p = 0.2961; *Figure 1C–D*). In order to test whether this protective effect of Cp was TrpV1 dependent, the TrpV1 antagonist capsazepine (Cz, 10 µM) was used. Gamma oscillation power was significantly reduced in hippocampal slices incubated with Cz together with Aβ and Cp compared to control (Aβ+Cz, power: 4.2 ± 0.92×$10^{-9}$ V$^2$, n = 7, p = 0.0122 vs control, *Figure 1C–D*) indicating that TrpV1 receptor activation mediates the protective properties of Cp. Similarly, gamma oscillation power was significantly reduced when hippocampal slices were incubated with Aβ and only Cz (Aβ+Cz, power: 2.2 ± 0.38×$10^{-9}$ V$^2$, n = 14, p < 0.0001 vs control, *Figure 1C–D*).

## TrpV1 receptor activation prevents Aβ-induced reduction of gamma oscillation rhythmicity

To characterize the preventative effect of TrpV1 activation on Aβ-induced degradation of hippocampal gamma oscillations, we first calculated the coefficient of rhythmicity (Cr) as a measure of the quality of gamma oscillations (see Materials and methods; *Cangiano and Grillner, 2003*). Concomitantly with decreasing gamma power, 50 nM Aβ significantly reduced Cr compared to control (control Cr: 0.50 ± 0.01, Aβ Cr: 0.45 ± 0.02, p = 0.0149, *Figure 1F–G*). This loss of rhythmicity caused by Aβ suggests an alteration in the synchronization of hippocampal cellular activity, which correlates with our previous findings (*Kurudenkandy et al., 2014*). Subsequent Cp treatment prevented the Aβ-induced reduction of Cr (Aβ+Cp Cr: 0.50 ± 0.02, p = 0.4003 vs control, *Figure 1F–G*) in a TrpV1-dependent manner since addition of Cz blocked the Cp protective effect (Aβ+Cp+Cz Cr: 0.42 ± 0.02, p = 0.0061 vs control, *Figure 1F–G*). These data suggest that TrpV1 receptor activation prevents Aβ-induced reduction of gamma oscillations through a mechanism affecting the synchrony of hippocampal cellular activity. Interestingly, Aβ together with Cz did not affect gamma rhythmicity since no reduction in Cr was found (Aβ+Cz Cr: 0.50 ± 0.03, p = 0.4572 vs control, *Figure 1G*).

## Capsaicin activates a TrpV1-dependent and a TrpV1-independent pathway

To control for possible effects of Cp on gamma oscillations in the absence of Aβ, hippocampal slices were incubated with 10 µM Cp for 15 min followed by bath perfusion of 100 nM KA to induce gamma oscillations. Interestingly, we found than Cp treatment alone induced a 66.5 ± 7.3% reduction of gamma power compared to control (control: 17.5 ± 3.1×10$^{-9}$ V$^2$, n = 14; Cp: 5.9 ± 1.3×10$^{-9}$ V$^2$, n = 7; p = 0.005; *Figure 1—figure supplement 1*). To determine if this inhibitory effect of Cp was mediated via activation of TrpV1 receptors, we incubated hippocampal slices with 10 µM Cp and 10 µM Cz (15 min). The resulting gamma oscillations remained depressed and were not significantly different from the results obtained with Cp alone (63.2 ± 7.0% reduction, Cp+Cz, power: 6.4 ± 1.2×10$^{-9}$ V$^2$, n = 15, p = 0.0014 vs control, *Figure 1—figure supplement 1*). Our data show that Cp activates two different, independent pathways: a TrpV1-dependent one, which mediates the protective effect of Cp against Aβ-induced reduction of gamma oscillations, and a TrpV1-independent one, which mediates the reduction of gamma oscillations caused by Cp itself. It is also apparent that the effect of Cp via the TrpV1-dependend pathway occludes its TrpV1-independent action in the presence of Aβ.

Because of this dual effect of Cp, we decided to further test the preventive effect of TrpV1 activation on Aβ-induced gamma oscillation impairment. To do this, we used hippocampal slices from TrpV1 knockout mice (TrpV1 KO). As shown in *Figure 2A–D*, slices treated with 50 nM Aβ (15 min) displayed a 51.2 ± 11.1% reduction in gamma power compared to control slices (control: 3.5 ± 0.7×10$^{-9}$ V$^2$, n = 10; Aβ: 1.7 ± 0.4×10$^{-9}$ V$^2$, n = 9; p = 0.0394). This reduction was similar to the one observed in WT slices (59.4 ± 8.9%, *Figure 1E*). As expected 10 µM Cp treatment did not prevent the Aβ-induced reduction of gamma power (Aβ+Cp: 0.9 ± 0.4×10$^{-9}$ V$^2$, n = 8; p = 0.0022 vs control, *Figure 2A–D*) supporting the above results on mediation of the Cp preventive effect via a TrpV1-dependent mechanism. Interestingly, TrpV1 KO slices incubated with Cp and Aβ displayed a greater reduction in gamma oscillations power (73.8 ± 11.8%) compared to Aβ-only incubation (Aβ: 1.7 ± 0.4×10$^{-9}$ V$^2$, n = 9; Aβ+Cp: 0.9 ± 0.4×10$^{-9}$ V$^2$, n = 8; p = 0.0296; *Figure 2A–D*). It is likely that this further reduction in gamma power is due to the TrpV1-independent effect of Cp observed in WT slices (*Figure 1—figure supplement 1*), which in the absence of the receptor in TrpV1 KO slices is no longer occluded.

In contrast to the Aβ effect on Cr observed in WT slices (*Figure 1G*), we found that Aβ did not reduce the quality of gamma oscillations in TrpV1 KO hippocampal slices (*Figure 2B*) since no changes of Cr were observed in Aβ-treated slices (control Cr: 0.49 ± 0.05, Aβ Cr: 0.51 ± 0.04, p = 0.4211, *Figure 2B,D*). Similar results were found in slices treated with Aβ and Cp (Aβ+Cp Cr: 0.42 ± 0.06, p = 0.1185 vs control, *Figure 2B–D*). This suggests that there are subtle differences in gamma oscillation characteristics in TrpV1 KO vs WT mice, which are outside the focus of this study.

## Desynchronization of pyramidal cell action potentials and shift of excitatory/inhibitory current balance are responsible for impairment of functional network dynamics by Aβ

Our above data on functional network dynamics provides evidence that Cp prevents Aβ-induced impairment of gamma oscillations in a TrpV1-dependent manner. Next, we proceeded to study the cellular and synaptic mechanisms underlying the TrpV1 preventive effect. As shown above, Cp prevented the Aβ-induced loss in gamma rhythmicity suggesting a mechanism modulating the synchronization of hippocampal neuronal activity. We explored this possibility by performing whole-cell patch-clamp recordings of pyramidal cells (PC) concomitant with LFP in a submerged recording chamber. Hippocampal slices were activated with 100 nM KA to induce gamma oscillations. For these experiments, Aβ was applied acutely to the bath and its concentration increased to 1 µM in order to offset the method-dependent lower signal amplitude in submerged conditions (*Kurudenkandy et al., 2014*). As shown in *Figure 3A*, 1 µM Aβ application for 15–20 min induced a 40.2 ± 7.02% reduction in gamma oscillation power (control: 8.6 ± 1.5×10$^{-10}$ V$^2$, Aβ: 5.2 ± 1.1×10$^{-10}$ V$^2$, n = 11, p = 0.0005, *Figure 3A,D*). In parallel with gamma power impairment Aβ also induced a 47.2 ± 10.1% reduction in AP firing frequency of PC (control: 2.0 ± 0.5 Hz, Aβ: 0.9 ± 0.2 Hz, n = 10, p = 0.0068, *Figure 3A,D*). No change in the membrane potential was observed

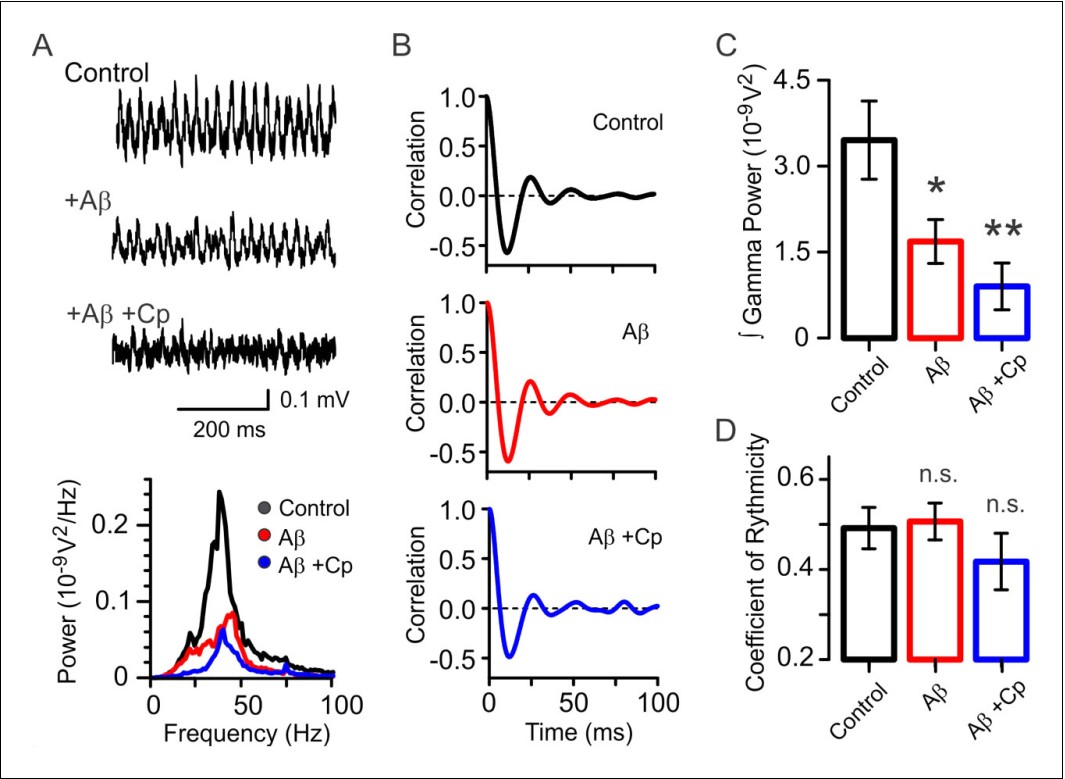

**Figure 2.** Capsaicin's preventative effect is absent in hippocampal slices from TrpV1 KO mice. (**A**) Representative sample traces and power spectra of KA-induced gamma oscillations in hippocampal slices from TrpV1 KO mice showing that Cp does not display the same protective properties against Aβ-induced reduction in gamma oscillations as in WT slices (*Figure 1C–G*). (**B**) Likewise, the coefficient of rhythmicity (Cr) remains unaltered after Aβ and Aβ+Cp treatments compared to control. (**C**) Integrated power of gamma oscillations from the experimental conditions described in A: Control (black), Aβ (red) and Aβ+Cp (blue). (**D**) Summary bar-graph of Cr from the experimental conditions described in B. Quantifications of integrated power and Cr were performed from the same 60s-segment 20 min after gamma oscillations induction by bath application of 100 nM KA (for experimental data see *Figure 2—source data 1*). Mann-Whitney test (one-tailed) was used for statistical significance on absolute values. Data is presented as mean ± SEM. * indicates p < 0.05, ** p < 0.01 and n.s. indicates no statistically significant differences.

DOI: https://doi.org/10.7554/eLife.37703.006
The following source data is available for figure 2:

**Source data 1.** Experimental data for *Figure 2*.
DOI: https://doi.org/10.7554/eLife.37703.007

after Aβ application compared to control (control: −40.4 ± 0.9 mV, Aβ: −40.1 ± 1.6 mV, n = 7, p = 0.3438, *Figure 3D*).

To determine the synchronization level of PC activity during gamma oscillations, spike-phase coupling was calculated before and after Aβ treatment (see Materials and methods). The results show that Aβ application slightly changed the AP preferred phase-angle (n = 9, control: 5.07 ± 0.02 radians, Aβ: 5.40 ± 0.04 radians, p < 0.0001, *Figure 3C,D*), which can be observed as a small shift to the right in the AP-phase distribution (*Figure 3C*). It is, however, open to question whether such a small phase angle change has any functional significance. More importantly, Aβ induced a desynchronization of PC activity that can be observed as an increase in the AP-phase Gaussian half-width of the AP firing window (control: 3.40 ± 0.62 radians, Aβ: 4.43 ± 0.79 radians, n = 9, p = 0.0098, *Figure 3B–D*). Our data corroborates our previous findings, which demonstrated that Aβ-induced reduction of gamma oscillations is caused by a desynchronization of AP firing in CA3 PC (*Kurudenkandy et al., 2014*).

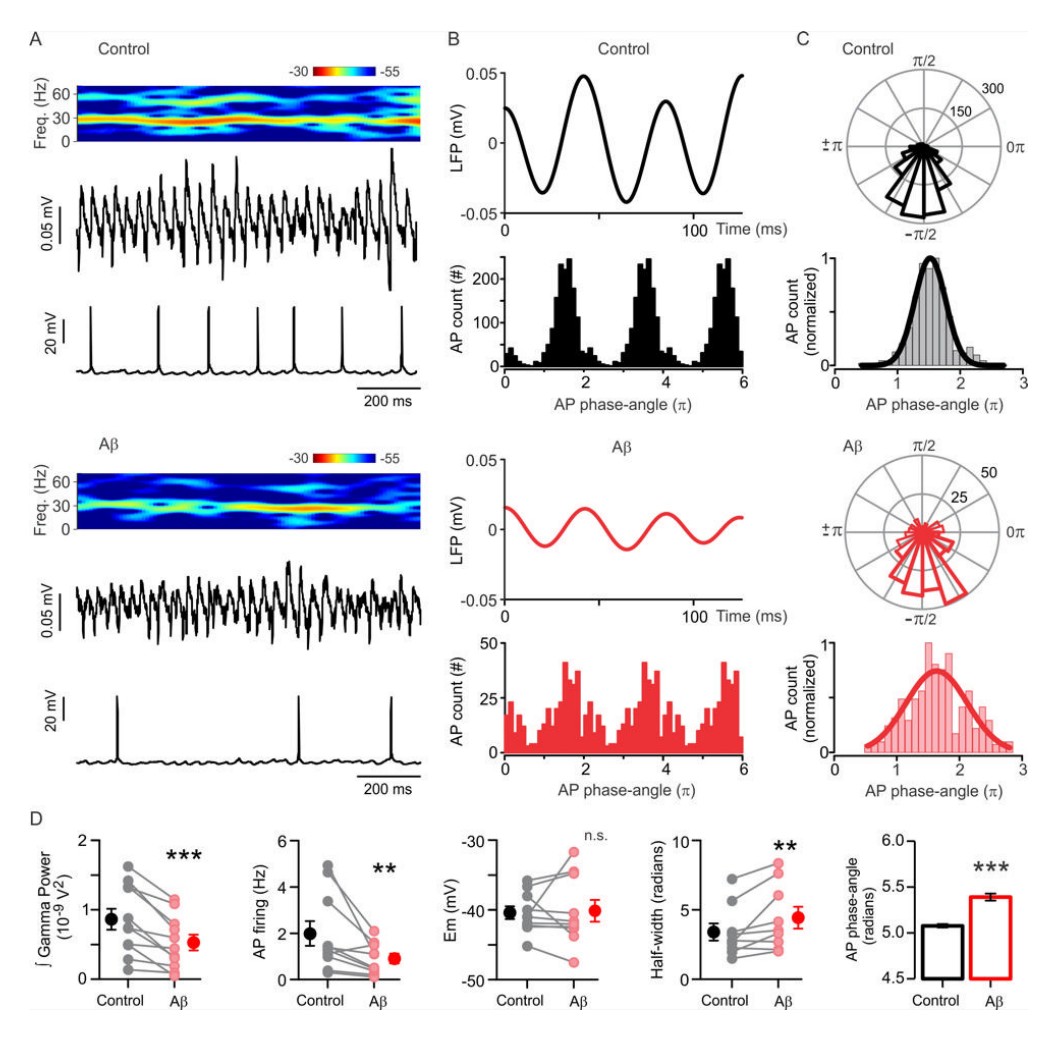

**Figure 3.** Aβ-induced desynchronization of AP firing in CA3 pyramidal cells. (**A**) Representative sample traces of concomitant LFP and patch-clamp recordings showing the Aβ-induced impairment of gamma oscillation power and AP firing rate. The spectrogram of the LFP activity shows the reduction in gamma oscillations power. (**B**) Filtered LFP recording showing three cycles of gamma oscillations (top) and frequency distribution plot of the AP gamma-phase-angles (bottom) in control (black) and after Aβ application (red). The frequency distribution plot shows the AP gamma-phase-angles repeated three times to represent the AP fire windows during gamma oscillations. (**C**) Representative polar-plots (top) showing the distribution of the AP phase-angles before (black) and after 1 μM Aβ application (red). Phase-angles and gamma oscillation-phases are presented in radians; the peak of the oscillation cycle corresponds to 0π and the trough corresponds to ±π. Normalized frequency-distribution (bottom) showing the AP phase-angles to which a Gaussian function was fitted to calculate the half-width as a measure of the synchronization level before and after Aβ application (see Materias and methods). Note that Aβ application induced a reduction in the number of AP (see AP count in (**B**) and polar plots scale in (**C**) and increased the AP fire window half-width. (**D**) Quantification of the integrated power, AP firing, AP firing window half-width and AP phase-angles before (control, black) and after 1 μM Aβ application (red). Note that Aβ induced a shift in the preferred phase-angle and a desynchronization of AP firing represented by an increase in the half-width of the AP firing windows (Gaussian curve fitted). Quantifications, except for half-width, were performed by averaging 5 min of control and the last 5 min of Aβ treatment with values taken over 1 min period (for experimental data see *Figure 3—source data 1*). Wilcoxon matched-pairs test (one-tailed) was used for statistical significance on absolute values in all quantifications except AP phase-angles for which statistically significant differences were tested by using the Mann-Whitney test (one-tailed). In all quantifications except for AP phase-angles each experiment is presented as before-after data and the mean ± SEM is also shown. ** indicates p < 0.01, ***p < 0.001 and n.s. indicates no statistically significant differences.

DOI: https://doi.org/10.7554/eLife.37703.008

*Figure 3 continued*

The following source data is available for figure 3:

**Source data 1.** Experimental data for *Figure 3*.

DOI: https://doi.org/10.7554/eLife.37703.009

Interestingly, the reduction in AP firing frequency found here is in contrast with our previous study in which Aβ induced an increase of AP firing in PC recorded in the cell-attached configuration (*Kurudenkandy et al., 2014*). In order to investigate whether this difference in Aβ effect could be explained by the different recording configuration employed here (whole-cell patch), we recorded AP as unitary events extracellularly and found that under these conditions Aβ application increased AP firing frequency as previously reported (control: 3.8 ± 0.9 Hz, Aβ: 4.9 ± 1.6 Hz, n = 5, p = 0.0313, for experimental data see *Source data 1*). For an explanation of this recording configuration-dependent difference see Discussion.

## TrpV1 receptor activation prevents Aβ-induced desynchronization of pyramidal cell action potentials

Next, we investigated whether TrpV1 receptor activation was able to prevent the Aβ-impairment of the cellular mechanisms responsible for the generation of gamma oscillations. We perfused slices with 1μM Aβ together with 10 μM Cp and, surprisingly, found a 34.7 ± 9.4% decrease in gamma oscillation power (control: 2.3 ± 1.1×10$^{-10}$ V$^2$, Aβ+Cp 15 min: 1.1 ± 0.3×10$^{-10}$ V$^2$, n = 6, p = 0.0313, *Figure 4—figure supplement 1*); similar to the reduction caused by 1 μM Aβ application alone (40.2 ± 7.02% of inhibition, *Figure 3A,D*). To test for a possible delayed action of Cp, we extended the treatment from 15 to 30 min. However, the reduction in gamma oscillation power persisted and even increased (Aβ+Cp 30 min: 42.6 ± 17.1% inhibition, power: 0.8 ± 0.3×10$^{-10}$ V$^2$, p = 0.0313 vs control, *Figure 4—figure supplement 1*). This increased gamma oscillation impairment was not evident in our previous study in which Aβ-induced reduction of gamma power remained stable after 15 min Aβ application (*Kurudenkandy et al., 2014*). It is, however, reminiscent of the further reduction observed in the slices from the TrpV1 KO mice (*Figure 2D*), suggesting that this increased impairment of gamma power might be due to the TrpV1-independent effect of Cp.

Since increasing the concentration of Cp may cause unspecific effects (*Benninger et al., 2008*; *Boudaka et al., 2007*; *Cao et al., 2007*; *Lundbaek et al., 2005*; *Yang et al., 2014*) and in order to avoid the inhibitory TrpV1-independent effect of Cp on gamma oscillations described above we decided to reduce Cp concentration to 5 μM. Our results show that this lower Cp concentration prevented the Aβ-induced impairment of gamma oscillation power (control: 2.8 ± 1.0×10$^{-10}$ V$^2$, Aβ+Cp: 2.6 ± 1.0×10$^{-10}$ V$^2$, n = 8, p = 0.3203, *Figure 4A,D*) as well as the reduction of AP firing rate (control: 0.86 ± 0.24 Hz, Aβ+Cp: 0.58 ± 0.14 Hz, n = 6, p = 0.2188, *Figure 4A,D*). In contrast, 5 μM Cp treatment did not prevent Aβ-induced effects in slices from the TrpV1 KO mice: neither on gamma power (control: 1.2 ± 0.4×10$^{-10}$ V$^2$, Aβ+Cp: 0.6 ± 0.4×10$^{-10}$ V$^2$, n = 6, p = 0.0156, *Figure 4E,H*) nor on AP firing (control: 1.30 ± 0.47 Hz, Aβ+Cp: 0.17 ± 0.06 Hz, n = 6, p = 0.0156, *Figure 4E,H*). This indicates that Cp preventive effect at 5 μM is reliant on TrpV1 receptor activation. After combined Aβ and Cp treatment, no change in membrane potential was observed in KA-activated slices of either WT (control: −43.5 ± 1.0 mV, Aβ+Cp: −42.2 ± 2.1 mV, n = 6, p = 0.1563) or TrpV1 KO mice (control: −41.7 ± 1.4 mV, Aβ+Cp: −43.7 ± 2.0 mV, n = 6, p = 0.2188).

By analyzing AP spike-phase coupling we found that 5 μM Cp treatment completely prevented the Aβ-induced desynchronization of PC activity in WT slices (*Figure 4B–D*; n = 5). No changes were observed any longer in the preferred phase-angle (control: 5.46 ± 0.04 radians, Aβ+Cp: 5.42 ± 0.04 radians, p = 0.2222, *Figure 4D*) and the half-width of the PC fire window (control: 3.98 ± 0.71 radians, Aβ+Cp: 3.93 ± 0.91 radians, p = 0.4206, *Figure 4C,D*). Interestingly, despite the reduction in gamma oscillation power and AP firing observed in TrpV1 KO slices, the spike-phase coupling of AP did not display any change after combined Aβ and Cp application (*Figure 4F–H*, n = 6); neither in the preferred phase-angle (control: 5.49 ± 0.03 radians, Aβ+Cp: 5.37 ± 0.08 radians, p = 0.0958, *Figure 4G,H*) nor in the half-width of the PC fire window (control: 4.25 ± 0.53 radians, Aβ+Cp: 5.01 ± 1.03 radians, n = 0.2813, *Figure 4G–H*).

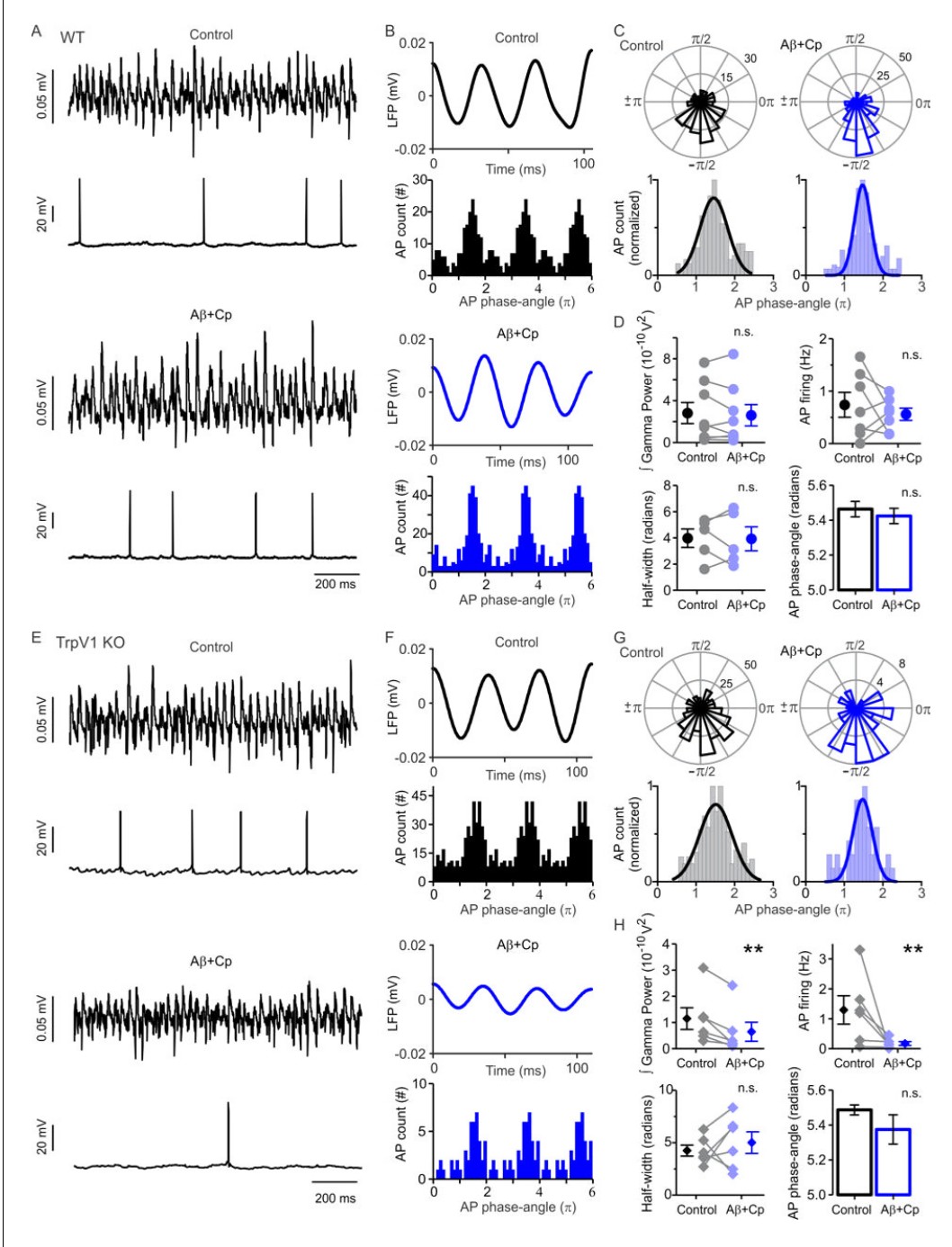

**Figure 4.** TrpV1 activation prevents Aβ-induced desynchronization of AP firing. (**A**) Representative sample traces of concomitant LFP and patch-clamp recordings in WT slices before (control, black) and after 1 μM Aβ plus 5 μM Cp application (blue). (**B**) Filtered WT LFP recording showing three cycles of gamma oscillations (top) and frequency distribution plot of the AP gamma-phase-angles (bottom) in control (black) and after Aβ+Cp application (blue). The frequency distribution plot shows the WT AP gamma-phase-angles repeated three times to represent the AP fire windows during gamma oscillations. (**C**) Representative polar-plots (top) showing the distribution of the WT AP phase-angles before (control, black) and after Aβ+Cp application (blue). Phase-angles and gamma oscillation-phases are presented in radians; the peak of the oscillation cycle corresponds to 0π and the trough corresponds to ±π. Normalized frequency-distribution (bottom) showing the AP phase-angles to which a Gaussian function was fitted to calculate the half-width to show the synchronization level before and after Aβ application (see Materials and methods). (**D**) Quantification of the integrated gamma power, AP firing, AP firing window half-

*Figure 4 continued on next page*

*Figure 4 continued*

width and AP phase-angles before (control, black) and after Aβ+Cp application (blue) in WT slices. (E) Representative sample traces of concomitant LFP and patch-clamp recordings in TrpV1-KO slices before (control, black) and after 1 µM Aβ plus 5 µM Cp application (blue). (F) Filtered TrpV1-KO LFP recording showing three cycles of gamma oscillations (top) and frequency distribution plot of the AP gamma-phase-angles (bottom) in control (black) and after Aβ+Cp application (blue). (G) Representative polar-plots (top) showing the distribution of the TrpV1-KO AP phase-angles before (control, black) and after Aβ+Cp application (blue). Normalized frequency-distribution (bottom) showing the AP phase-angles with the Gaussian function fitted. (H) Quantification of the integrated gamma power, AP firing, AP firing window half-width and AP phase-angles before (control, black) and after Aβ+Cp application (blue) in TrpV1-KO slices. Note that application of 5 µM Cp together with 1 µM Aβ prevented the Aβ-related reduction in KA-induced gamma oscillation power and AP firing rate in WT but not in TrpV1-KO slices demonstrating that the preventive role of Cp is mediated by the activation of TrpV1. Quantifications, except for half-width, were performed by averaging 5 min of control and the last 5 min of Aβ+Cp treatment with values taken over 1 min period. Wilcoxon matched-pairs test (one-tailed) was used for statistical significance on absolute values in all quantifications except AP phase-angles for which statistically significant differences were tested by using the Mann-Whitney test (one-tailed). In all quantifications except for AP phase-angles each experiment is presented as before-after data and the mean ± SEM is also shown. ** indicates p < 0.01 and n.s. indicates no statistically significant differences (for experimental data see *Figure 4—source data 1*).
DOI: https://doi.org/10.7554/eLife.37703.010

The following source data and figure supplements are available for figure 4:

**Source data 1.** Experimental data for *Figure 4*.
DOI: https://doi.org/10.7554/eLife.37703.013
**Figure supplement 1.** 10 µM capsaicin does not prevent Aβ-induced reduction in gamma oscillation power.
DOI: https://doi.org/10.7554/eLife.37703.011
**Figure supplement 1—source data 1.** Experimental data for *Figure 4—figure supplement 1*.
DOI: https://doi.org/10.7554/eLife.37703.012

This result suggests that the mechanism underlying the reduction of gamma oscillation power observed in TrpV1 KO slices is related to the decrease in PC activity. It is in accordance with our previous observation that gamma oscillations show no Cr change in TrpV1 KO slices incubated with Aβ (*Figure 2D*). In summary, our data suggest that a low Cp concentration (5 µM) reverses Aβ-induced desynchronization of PC activity via a TrpV1-dependent mechanism.

## TrpV1 receptor activation prevents Aβ-induced shift of excitatory/inhibitory current balance

Previously, we demonstrated that Aβ-induced impairment of gamma oscillations was, together with AP desynchronization of PC, caused by a shift in the balance of excitatory (EPSC) and inhibitory (IPSC) postsynaptic currents in CA3 PC (*Kurudenkandy et al., 2014*). To test whether Cp treatment could prevent this shift, we recorded EPSC ($V_h$ = -70mV) and IPSC ($V_h$ = 0 mV) in hippocampal PC in slices activated with 100 nM KA (20 min) in a submerged recording chamber. For these experiments, we used 1 µM Aβ together with 5 µM Cp since we showed above that at this Cp concentration causes TrpV1-dependent effects. As shown in *Figure 5*, 1 µM Aβ application induced a 22.9 +/- 9.5% reduction in EPSC amplitude (control: 26.7 ± 1.8 pA, Aβ: 21.0 ± 3.3 pA, n = 7, p = 0.0391, *Figure 5A*) while no effect was observed on EPCS frequency (control: 34.7 ± 2.3 Hz, Aβ: 36.7 ± 3.3 Hz, n = 7, p = 0.2891, *Figure 5A*) and charge transfer (control: 490.8 ± 50.8 pC, Aβ: 433.7 ± 82.6 pC, n = 7, p = 0.1875, *Figure 5A*). The Aβ effect on EPSC amplitude was prevented by the application of Cp since no remaining reduction in EPSC amplitude was found (control: 21.1 ± 3.1 pA, Aβ +Cp: 21.2 ± 4.8 pA, p = 0.3125, *Figure 5B*). No changes were observed for EPSC frequency with application of Cp together with Aβ (control: 35.1 ± 7.4 Hz, Aβ+Cp: 34.1 ± 10.6 Hz, n = 5, p = 0.5, *Figure 5B*) as well as for charge transfer (control: 88.0 ± 18.4 pC, Aβ+Cp: 92.5 ± 34.6 pC, n = 5, p = 0.5, *Figure 5B*). As expected, the co-application of Aβ and Cp together with the TrpV1 receptor antagonist Cz (Aβ+Cp+Cz) blocked Cp's preventive effects since we observed a 18.8 +/- 5.1% reduction in EPSC amplitude (control: 19.8 ± 2.7 pA, Aβ+Cp+Cz: 15.9 ± 2.3 pA, n = 8, p = 0.0039, *Figure 5C*), indicating that TrpV1 receptor activation is mediating the preventive effect of Cp against the Aβ-induced effects. No changes effected by Aβ+Cp+Cz treatment were observe for EPSC

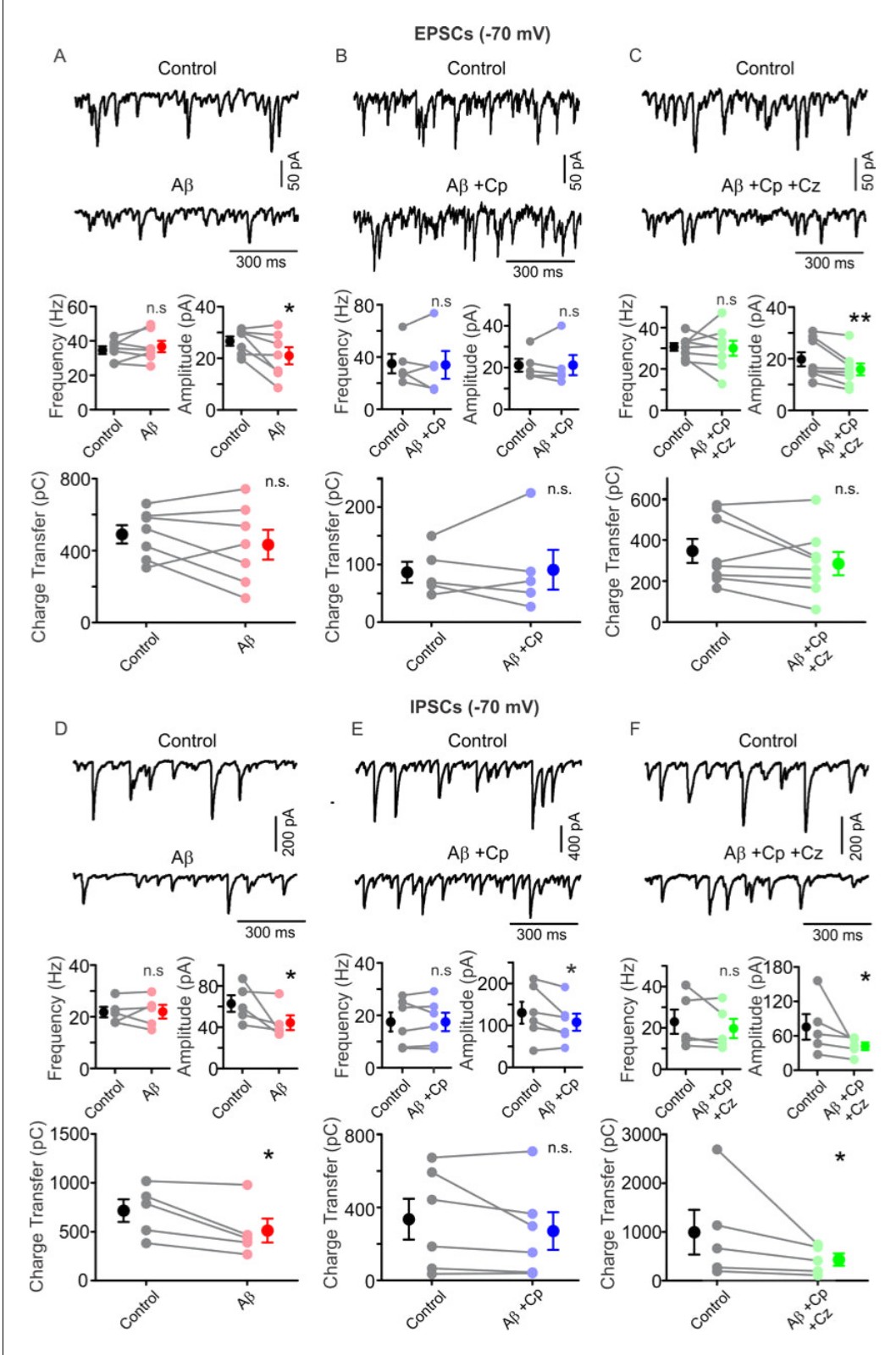

**Figure 5.** TrpV1 activation prevents Aβ-induced impairment of EPSC/IPSC current balance. (**A**) Representative traces (top) of EPSCs (PC, Vh=-70mV) before (control, black) and after 1 μM Aβ application (red) in hippocampal slices activated with 100 nM KA. Summary-graph (bottom) of the EPSCs frequency, amplitude and charge transfer before (control, black) and after Aβ application (red). (**B**) Representative traces (top) of EPSCs (PC, Vh=-70mV)

*Figure 5 continued on next page*

*Figure 5 continued*

before (control, black) and after 1 μM Aβ plus 5 μM Cp application (blue) in hippocampal slices activated with 100 nM KA. Summary-graph (bottom) of the EPSC frequency, amplitude and charge transfer before (control, black) and after Aβ+Cp application (blue). (C) Representative traces (top) of EPSCs (PC, Vh=-70mV) before (control, black) and after 10 μM Cz together with 1 μM Aβ plus 5 μM Cp application (green) in hippocampal slices activated with 100 nM KA. Summary-graph (bottom) of the EPSC frequency, amplitude and charge transfer before (control, black) and after Aβ+Cp+ Cz application (green). (D) Representative traces (top) of IPSCs (PC, Vh=-70mV) before (control, black) and after 1 μM Aβ application (red) in hippocampal slices activated with 100 nM KA. Summary-graph (bottom) of the IPSC frequency, amplitude and charge transfer before (control, black) and after Aβ application (red). (E) Representative traces (top) of IPSCs (PC, Vh=-70mV) before (control, black) and after 1 μM Aβ plus 5 μM Cp application (blue) in hippocampal slices activated with 100 nM KA. Summary-graph (bottom) of the IPSC frequency, amplitude and charge transfer before (control, black) and after Aβ+Cp application (blue). (F) Representative traces (top) of IPSCs (PC, Vh=-70mV) before (control, black) and after 10 μM Cz together with 1 μM Aβ plus 5 μM Cp application (green) in hippocampal slices activated with 100 nM KA. Summary-graph (bottom) of the IPSC frequency, amplitude and charge transfer before (control, black) and after Aβ+Cp+ Cz application (green). Note that Cp treatment reversed Aβ-related impairment of excitatory/inhibitory current balance by restoring EPSC amplitude and IPSC charge transfer. Quantifications were performed by averaging 5 min of control and the last 5 min of treatment with values taken over 1 min period (for experimental data see *Figure 5—source data 1*). Wilcoxon matched-pairs test (one-tailed) was used for statistical significance on absolute values. Each experiment is presented as before-after data and the mean ±SEM is shown. * indicates $p < 0.05$, **$p < 0.01$ and n.s. indicates no statistically significant differences.

DOI: https://doi.org/10.7554/eLife.37703.014

The following source data and figure supplements are available for figure 5:

**Source data 1.** Experimental data for *Figure 5*.
DOI: https://doi.org/10.7554/eLife.37703.017
**Figure supplement 1.** Cp does not prevent the Aβ-induced impairment of IPSC parameters recorded at Vh = 0 mV.
DOI: https://doi.org/10.7554/eLife.37703.015
**Figure supplement 1—source data 1.** Experimental data for *Figure 5—figure supplement 1*.
DOI: https://doi.org/10.7554/eLife.37703.016

frequency (control: 30.7 ± 1.8 Hz, Aβ+Cp+Cz: 30.0 ± 3.6 Hz, n = 8, p = 0.3203, *Figure 5C*) and charge transfer (control: 350.0 ± 58.3 pC, Aβ+Cp+Cz: 288.8 ± 56.5 pC, n = 8, p = 0.0977, *Figure 5C*).

Next, we investigated the effect of Cp treatment on the Aβ-related alterations of IPSCs. Similarly to EPSCs, 1 μM Aβ induced a 34.4 ± 5.4% reduction of IPSC amplitude (control: 88.2 ± 15.8 pA, Aβ: 57.7 ± 10.3 pA, n = 9, p = 0.002, *Figure 5—figure supplement 1A*). Moreover, Aβ also induced a 32.5 ± 6.5% reduction of IPSC charge transfer (control: 1073.0 ± 183.8 pC, Aβ: 691.7 ± 101.1 pC, n = 9, p = 0.002, *Figure 5—figure supplement 1A*). No alterations following Aβ application was found on IPSC frequency (control: 23.9 ± 0.7 Hz, Aβ: 24.1 ± 1.0 Hz, n = 9, p = 0.2480, *Figure 5—figure supplement 1A*). In contrast to EPSCs, Cp did not prevent Aβ-induced alterations of IPSC parameters. Our experiments evidenced a persistent 23.9 ± 9.4% reduction in IPSC amplitude (control: 71.37 ± 13.5 pA, Aβ+Cp: 55.4 ± 14.3 pA, n = 6, p = 0.0156, *Figure 5—figure supplement 1B*) and a 30.3 ± 10.3% reduction in charge transfer (control: 1214 ± 238.8 pC, Aβ+Cp: 858.9 ± 205.1 pC, n = 6, p = 0.0156, *Figure 5—figure supplement 1B*). No changes in IPSC frequency were observed (control: 31.5 ± 2.4 Hz, Aβ+Cp: 29.5 ± 3.0 Hz, n = 6, p = 0.1094, *Figure 5—figure supplement 1B*).

The TrpV1 receptor is a non-selective ion channel with high permeability to $Ca^{2+}$(*Caterina et al., 1997*). Since the holding potential used to record the IPSC ($V_h$ = 0 mV) was similar to the TrpV1 receptor reversal potential when activated by Cp (*Caterina et al., 1997*), we decided to perform a second series of experiments with a high-chloride intracellular solution in order to shift TrpV1 reversal potential away from the recording $V_h$ (−70 mV). To control for possible differences in the Aβ-related effects by recording the IPSC under these conditions, we performed experiments with Aβ application alone. The results showed no differences in the Aβ-effect on IPSCs since a 26.8 ± 10.4% reduction was found in amplitude (control: 62.9 ± 8.0 pA, Aβ: 44.4 ± 7.2 pA, n = 5, p = 0.0313, *Figure 5D*) and a 29.7 ± 7.7% reduction in charge transfer (control: 713.0 ± 115.8 pC, Aβ:

507.8 ± 122.7 pC, n = 5, p = 0.0313, *Figure 5D*). Similarly to prior experiments, no changes were observed in IPSC frequency (control: 21.8 ± 2.0 Hz, Aβ: 22.0 ± 2.6 Hz, n = 5, p = 0.5, *Figure 5D*).

Nonetheless, under these conditions, we found that 5 µM Cp completely prevented the Aβ-induced reduction in IPSC charge transfer (control: 333.0 ± 112.0 pC, Aβ+Cp: 268.1 ± 103.3 pC, n = 6, p = 0.1563, *Figure 5E*). In addition, Cp partially prevented the Aβ-effect on amplitude since only a 12.7 ± 7.5% reduction was observed (control: 130.3 ± 26.1 pA, Aβ+Cp: 107.7 ± 20.4 pA, n = 6, p = 0.0313, *Figure 5E*), in contrast to the 26.8 ± 10.4% reduction found with Aβ application alone (*Figure 5D*). IPSC frequency remained unaltered (control: 17.4 ± 3.6 Hz, Aβ+Cp: 17.4 ± 3.5 Hz, n = 6, p = 0.2813, *Figure 5E*).

Finally, we performed experiments using the TrpV1 receptor antagonist Cz together with Aβ and Cp (Aβ+Cp+Cz). Our results demonstrate that Cp's preventive effect on IPSC parameters was mediated by TrpV1 receptor activation since we observed a 34.3 ± 10.6% reduction in IPSC amplitude (control: 75.4 ± 22.2 pA, Aβ+Cp+Cz: 41.8 ± 6.8 pA, n = 5, p = 0.0313, *Figure 5F*) and a 44.1 ± 7.5% reduction in charge transfer (control: 992.1 ± 457.3 pC, Aβ+Cp+Cz: 430.0 ± 126.8 pC, n = 5, p = 0.0313, *Figure 5F*). As expected, no changes in IPSC frequency were observed (control: 23.0 ± 5.9 Hz, Aβ+Cp+Cz: 19.7 ± 4.7 Hz, n = 5, p = 0.2188, *Figure 5F*). This suggests that Cp treatment recovers the inhibitory input to CA3 PC via a postsynaptic mechanism involving $Ca^{2+}$ influx through TrpV1 receptor activation since the Cp protective effect was absent when experiments were performed at $V_h$ = 0 mV. Taken together, our results suggest that Cp prevents the Aβ-induced shift in excitatory/inhibitory current balance in hippocampal gamma oscillations induced with KA.

## TrpV1 receptor activation rescues Aβ-induced impairment of functional network dynamics

In the context of clinical treatment possibilities, it is important to establish whether a potential therapeutic approach is able to just prevent pathological damage or, more desirably, is also capable of rescuing impairment after pathological damage has already occurred. We therefore proceeded to investigate whether the preventive effect of TrpV1 activation by Cp demonstrated above extend to restorative capabilities. Specifically, we asked whether Cp could rescue the Aβ-induced impairment of functional network dynamics in the form of gamma oscillations. To assess this, continuous recordings of LFP gamma oscillation activity were performed in four different experimental conditions (*Figure 6*). In three experimental groups, all slices were first incubated with 50 nM Aβ for 15 min and transferred to an interface recording chamber where gamma oscillations were induced by bath perfusion of 100 nM KA and allowed to stabilize for 20 min. The first Aβ-pre-incubated group was used to monitor, without any further treatment, the stability of the Aβ-impaired gamma oscillations. (Aβ, n = 18); the second Aβ-pre-incubated group was treated with bath application of 10 µM Cp (Aβ +Cp, n = 16); and the third Aβ-pre-incubated group was treated with 10 µM Cp plus 10 µM Cz to test for TrpV1-specificity (Aβ+Cp+Cz, n = 11). In addition, one group of slices was activated just with KA, without prior Aβ incubation, to monitor gamma oscillations under normal conditions (control, n = 15).

Compared to control gamma oscillation power (17.5 ± 2.8×$10^{-9}$ $V^2$, *Figure 6C–D*), hippocampal slices of the three groups pre-incubated with 50 nM Aβ displayed a comparable reduction in gamma power after 20 min KA and before further treatment application (Aβ: 7.0 ± 1.7×$10^{-9}$ $V^2$, p = 0.001 vs control; Aβ+Cp: 7.9 ± 1.5×$10^{-9}$ $V^2$, p = 0.0034 vs control; Aβ+Cp+Cz: 6.6 ± 1.0×$10^{-9}$ $V^2$, p = 0.0041 vs control; *Figure 6C*). Gamma oscillation power measurements after Cp treatment application (30 min) showed that Cp rescued impaired gamma power to levels similar to those recorded in control slices at the same time point (control 30 min: 22.1 ± 3.7×$10^{-9}$ $V^2$, Aβ+Cp 30 min: 18.7 ± 3.9×$10^{-9}$ $V^2$, p = 0.2323, *Figure 6A–D*). In contrast, slices incubated just with Aβ (with no Cp application), showed a persistently impaired gamma oscillation power (Aβ 30 min: 9.9 ± 2.1×$10^{-9}$ $V^2$, p = 0.0018 vs control, *Figure 6A–D*). Finally, we showed that the Cp rescue effect was TrpV1-dependent since the application of Cz together with Cp prevented the rescue of impaired gamma oscillation power (Aβ+Cp+ Cz 30 min: 11.1 ± 1.7×$10^{-9}$ $V^2$, p = 0.0074 vs control, *Figure 6A–D*).

Overall, our results suggest that TrpV1 activation is a promising therapeutic target for attempts to restore Aβ-impaired functional network dynamics and cellular mechanisms back to physiological levels. Whether this restoration of cognition-relevant network function translates to demonstrable cognitive benefit in vivo is the focus of ongoing studies.

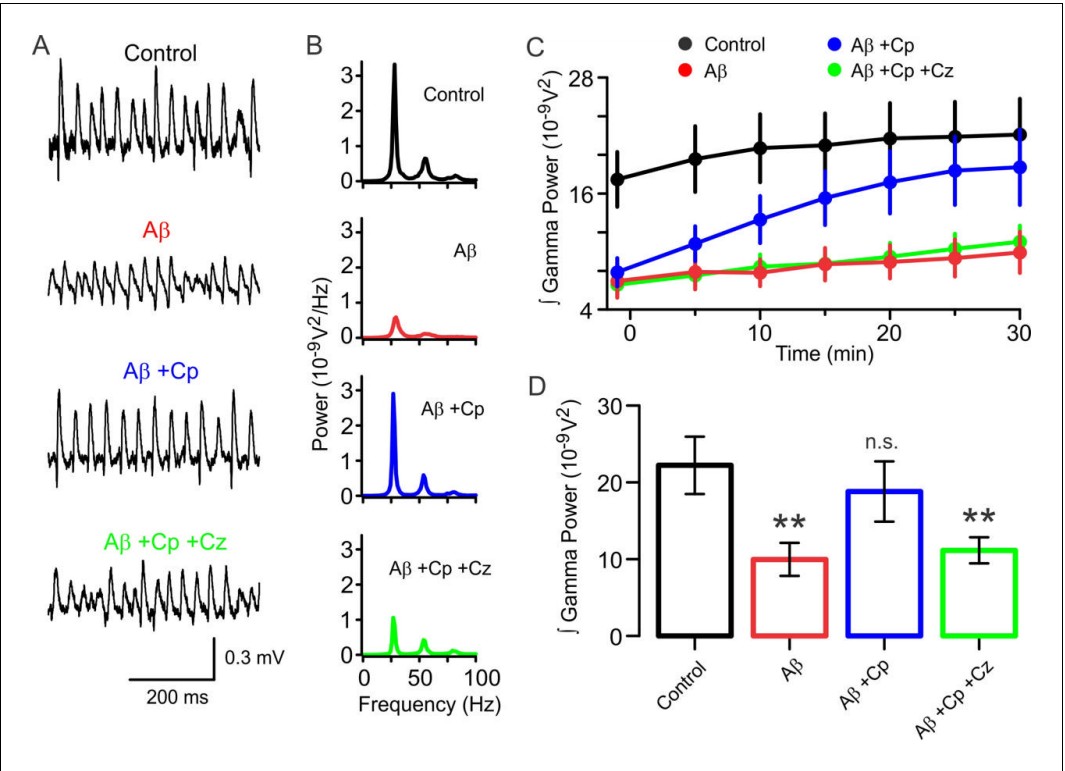

**Figure 6.** TrpV1 activation rescues the hippocampal network from Aβ-induced degradation of gamma oscillations. (A,B) Representative sample traces and power spectra of KA-induced gamma oscillations in hippocampal slices after 30 min of control activity (black), and from slices pre-incubated with 50 nM Aβ (Aβ, red). Gamma power from Aβ-pre-incubated slices treated with 10 µM Cp (Aβ+Cp, blue), and Aβ-pre-incubated slices treated with 10 µM Cp together with 10 µM Cz (Aβ+Cp+Cz, green). Note that Cp treatment rescues gamma oscillations to levels similar to control. (C) Time-course of the integrated power of gamma oscillations from the experimental conditions described in A and B showing the time-dependent increase in gamma power in the slices treated with Cp (blue). Note that gamma power in Aβ-pre-incubated slices not treated with Cp remained low (Aβ, red). Similar results were obtained when Cz was included with the treatment (Aβ+Cp+Cz, green) indicating that the TrpV1 receptor mediates the restorative effect exerted by Cp. (D) Summary bar-graph of the integrated gamma power from the experimental conditions described in A. Integrated power was measured on 60 s segments every 5 min after treatment application. Quantification was performed after 30 min of treatment application. In all experiments, recordings were started 20 min after gamma oscillation induction by 100 nM KA bath application (for experimental data see *Figure 6—source data 1*). Mann-Whitney test (one-tailed) was used for statistical significance on absolute values. Data is presented as mean ± SEM. * indicates p < 0.05 and n.s. indicates no statistically significant differences.

DOI: https://doi.org/10.7554/eLife.37703.018

The following source data is available for figure 6:

**Source data 1.** Experimental data for *Figure 6*.
DOI: https://doi.org/10.7554/eLife.37703.019

## Discussion

### TrpV1 in the central nervous system

Since its discovery in the dorsal root ganglia as the receptor for capsaicin (Cp; *Caterina et al., 1997*), knowledge of TrpV1 distribution in the central nervous system (CNS) has expanded and today is understood to include hippocampus (*Cristino et al., 2006*; *Mezey et al., 2000*; *Roberts et al., 2004*; *Sasamura et al., 1998*). While the function of TrpV1 in nociception and sensory transmission is well documented, its role in CNS is less well understood. Growing evidence suggests that TrpV1 is involved in several brain functions by modulating glial and neuronal activity in normal and pathological conditions (*Edwards, 2014*; *Ho et al., 2012*; *Martins et al., 2014*).

In this study, we found that cannabinoid receptor activation is unable to protect cognition-relevant functional network dynamics against Aβ-induced impairment. Conversely, TrpV1 activation by Cp displayed strong protective and restorative properties, preventing and rescuing the Aβ-induced degradation of hippocampal gamma oscillations and relevant cellular mechanisms. In contrast to the preventive role of TrpV1 activation found here, *Benito et al. (2012)* have reported that Aβ can induce a pathological inflammatory response in primary astrocyte cultures in which several membrane receptors, including TrpV1, are involved. These differing consequences of TrpV1 activation could be explained by the different Aβ preparations and concentrations used in both studies: our Aβ preparation contains a mix of different aggregation states (monomers, oligomers, fibrils; *Kurudenkandy et al., 2014*) while Benito and coworkers used a purely fibrillar Aβ preparation (*Benito et al., 2012*), which is more toxic in our experimental model (*Kurudenkandy et al., 2014*; *Cohen et al., 2015*). Also, we use physiologically relevant concentrations of recombinant Aβ (50 nM; *Roher et al., 2009*) for LFP recordings while Benito and coworkers use a very high 5 µM of synthetic Aβ. Since TrpV1 is known to play an important role in anti-inflammatory processes (*Devesa et al., 2011*; *Tsuji and Aono, 2012*) it is possible that the effect of 5 µM fibrillar Aβ induces a pathological activation of inflammatory mechanisms in the primary astrocytes culture. Hyperactivation of immune responses (neuroinflammation) induced by Aβ has also been shown in other studies (*Blasko et al., 2004*; *Cai et al., 2014*). Similarly to chronic inflammation, in which processes originally activated to restore system homeostasis become pathological due to an imbalance in the regulatory mechanisms (*Devesa et al., 2011*; *Medzhitov, 2008*), the 5 µM fibrillar Aβ used by Benito and coworkers could induce a pathological TrpV1 function.

On the other hand, similar to our study, TrpV1 activation as a therapeutic target has been tested in a mouse model of ischemia (*Cao et al., 2014*). Cao and coworkers found that TrpV1 activation leads to a decrease in the volume of affected brain tissue after the induction of cerebral ischemia. Furthermore, an improvement in a behavioral-neurofunction-test score was found in animals treated with Cp after brain ischemia. Another recent study has also reported that TrpV1 activation in astrocytes in a rat model of Parkinson's disease is involved in the prevention of dopamine neuron degeneration in substantia nigra (*Nam et al., 2015*). Moreover, TrpV1 activation in astrocytes also led to behavioral rescue in this rat PD model since a reduction in the pathologically induced rotational behavior was observed after Cp treatment (*Nam et al., 2015*). These data, together with the findings in this study, strongly suggest that TrpV1 receptor activation is a suitable therapeutic approach in the fight against neurodegenerative disorders and the cognitive decline associated with them.

## TrpV1 activation and gamma oscillations

Gamma oscillations emerge from synchronous postsynaptic currents and AP firing of different cell types as a result of the coordinated interaction between excitation and inhibition in a network (*Buzsáki and Wang, 2012*; *Mann et al., 2005*). Here, we found that TrpV1 activation by Cp prevented the reduction in AP firing rate of PCs and reversed the Aβ-induced increase in the AP gamma-phase firing window (desynchronization) suggesting that TrpV1 activation protects gamma oscillations by maintaining the synchrony of neuronal activity in hippocampus. Furthermore, our data show that TrpV1 activation protects the excitatory/inhibitory current balance against the Aβ-related impairment. Hence, it is possible that TrpV1 activation triggers synaptic mechanisms that offset the Aβ-induced shift of excitatory/inhibitory current balance and, consequently, sustains the synchrony of neuronal AP firing during hippocampal gamma oscillations. In support of this hypothesis, it has been shown that TrpV1 modulates excitatory transmission in CNS either presynaptically by modulating glutamate release (*Benson et al., 2000*; *Marinelli et al., 2002*; *Marinelli et al., 2003*; *Gibson et al., 2008*; *Musella et al., 2009*) or postsynaptically through the internalization of AMPA receptors (*Chávez et al., 2010*; *Grueter et al., 2010*). TrpV1 activation by Cp in hippocampus has been reported to depress EPSC and induce a form of long-term synaptic depression (LTD) in CA1 interneurons (*Gibson et al., 2008*) and granular cells in dentate gyrus (*Chávez et al., 2010*). These effects in excitatory transmission onto CA1 interneurons are related to a mechanism involving the presynaptic rise in $Ca^{2+}$, calcineurin activation and the subsequent depression of neurotransmission (*Jensen and Edwards, 2012*) while in the dentate gyrus the postsynaptic rise in $Ca^{2+}$ and calcineurin activation mediates a clathrin-dependent internalization of AMPA receptors (*Chávez et al., 2010*). In addition, we found that TrpV1 activation reverses the Aβ-induced impairment of IPSC. Furthermore, this protective effect of TrpV1 activation was absent when IPSC were recorded at a holding potential

near the TrpV1 reversal potential (0 mV) suggesting, firstly, that TrpV1 is located at the postsynaptic terminal and, secondly, that TrpV1 activation by Cp protects inhibitory transmission through a mechanism related to the $Ca^{2+}$ influx into the postsynaptic terminal. Taken together, the evidence cited above suggests that the regulation of synaptic function by TrpV1 receptor activation through the modulation of excitatory/inhibitory current balance on pyramidal neurons could be the mechanism by which Cp reverses the Aβ-induced desynchronization of AP firing and, thereby, protects hippocampal gamma oscillations.

## Aβ-induced neuronal metabolic dysfunction and choice of recording configuration

Another interesting finding of this study is that the Aβ-induced reduction of AP firing rate in PC of WT hippocampal slices is in contrast with our previously published results (*Kurudenkandy et al., 2014*), where Aβ induced an increase in AP firing rate in PC while the desynchronization of AP-phase coupling observed after Aβ treatment is the same in both studies. These seemingly contradictory Aβ effects are explained by the differing recording conditions employed for these experiments in the two studies: here AP firing activity was monitored in the whole-cell configuration while the recordings in our previous study were performed in cell-attached configuration. We tested this in the present study by performing additional unitary spike recordings and found that Aβ application under these conditions increased AP firing rate as we previously reported (*Kurudenkandy et al., 2014*). The differing results depending on recording configuration are explained by unpublished data from our lab showing that Aβ reduces intracellular ATP levels and, in parallel, induces a membrane potential depolarization in neurons recorded in the perforated-patch but not in neurons recorded in the whole-cell configuration. This ATP deficit can be compensated with ATP contained in the intracellular recording solution. Hence, the reduction in AP firing rate observed using whole-cell patch recordings in the present study is likely caused by the metabolic dysfunction induced by Aβ and its compensation by ATP diffusing into the cytosol from the intracellular pipette solution. However, further experiments are needed to study this possibility in more detail.

Also, by performing concomitant experiments in TrpV1 KO hippocampal slices, we found that despite the Aβ-induced reduction of AP firing and gamma oscillation power, the Aβ-induced changes to the preferred AP phase-angle and PC fire window were no longer observed after Cp treatment. This suggests that Aβ-induced degradation of gamma oscillations in these slices may be caused by a mechanism involving just the decrease in AP firing rate. These results can explain why no effect of Aβ exposure on the coefficient of rhythmicity was found in TrpV1 KO slices and also suggests that TrpV1 function in the knockout mice could be compensated for another member of the TRP receptor family.

## Pathology-selective activation of TrpV1-dependent vs TrpV1-independent mechanisms

Our data indicate that the TrpV1 preventive effect seems to be regulated by a mechanism triggered by Aβ since the protective effect was observed when Cp was applied together with Aβ. TrpV1 activation by Cp may only be possible under the pathological conditions created by Aβ while in physiological conditions a TrpV1-independent mechanism is activated. This hypothesis is supported by recent studies, which showed that TrpV1 expression is increased in hippocampal CA1 and CA3 pyramidal neurons of a rat model of epilepsy (*Saffarzadeh et al., 2015*; *Saffarzadeh et al., 2016*). Similarly, *Bhaskaran and Smith (2010)* found an increase in TrpV1 expression in the dentate gyrus of a mouse model of temporal lobe epilepsy. These studies indicate that TrpV1 receptor function can be upregulated in pathological conditions and supports the hypothesis that Aβ can trigger the activation of a mechanism involved in the regulation of TrpV1 expression and/or function. In light of this, a possible explanation for our data arises from a recent study in which high-glucose concentration in the culture medium of a PC12 cell line increased *TrpV1* mRNA levels (*Mohammadi-Farani et al., 2014*). Accordingly, it has been reported that Aβ-induced pathological effects include glucose uptake reduction (*Prapong et al., 2002*; *Uemura and Greenlee, 2001*), which concurs with our own unpublished data. Hence, in our experimental approach, extracellular glucose levels could be increased by the Aβ-induced reduction in glucose uptake and, thus, induce the expression of TrpV1 receptor.

This hypothesis can also explain the long-lasting effects of Cp we observed in this study despite the well-known desensitization of the TrpV1 receptor (*Ho et al., 2012*). Assuming that Aβ-driven increase in extracellular glucose concentration is constant, *TrpV1* mRNA expression could renovate the available pool of TrpV1 in the cell membrane. Similarly, since the rescue effect of Cp would depend on TrpV1 expression and translocation to the cell membrane, TrpV1 activation and its preventive effects would be expected to be time-dependent as observed in the slow Cp-dependent increase in gamma power displayed in the rescue experiments in the present study. Finally, as cited above, TrpV1 activation can induce a form of LTD on hippocampal interneurons (*Gibson et al., 2008*), indicating that TrpV1 physiological effects are long-lasting. Hence this mechanism could also explain the long-lasting protective effects of Cp we observed in this study.

Very limited data is available regarding TrpV1 receptor expression in the human central nervous system. Only Mezey and coworkers (2000) have reported that TrpV1 is expressed in the parietal cortex (*Mezey et al., 2000*). *Cavanaugh et al. (2011)* have reported no detectable TrpV1 expression in the human hippocampus (*Cavanaugh et al., 2011*). Although more studies assessing the expression of TrpV1 in the human brain under different conditions are needed, the lack of detectable TrpV1 expression in normal human hippocampus found by *Cavanaugh et al. (2011)* supports our hypothesis that TrpV1 expression could be up-regulated under pathological conditions only. Potential induction of TrpV1 receptor expression can be studied using AD animal models and/or brain slice assays and, more importantly, TrpV1 activation as a therapeutic target can be assessed further.

Support for the TrpV1-independent mechanism activated by Cp reported here can be found in a study that shows Cp inducing a reduction in the amplitude of evoked EPSC in granule cells of the dentate gyrus in both WT and TrpV1 KO mice (*Benninger et al., 2008*). Moreover, this Cp effect on excitatory transmission seems to be related to a reduction in glutamate release from presynaptic terminals since a change in the paired-pulse ratio was observed with similar proportions in WT and TrpV1 KO mice. In our study, the activation of this TrpV1-independent mechanism was related to a reduction in gamma oscillation power. One possibility to explain this effect is the property of Cp to regulate the excitability of neurons through modulation of ion channel activity. Supporting this hypothesis, *Cao et al. (2007)* have reported that Cp can regulate voltage-gated sodium channel (VGSC) activity in a TrpV1-independent manner. They found that high concentrations of Cp (>10 μM) induced a reduction in the amplitude of VGSC currents and shifted the inactivation curve to more negative potentials (*Cao et al., 2007*). Similarly, *Yang et al. (2014)* have found that Cp can regulate voltage-gated potassium channels (VGPC) responsible for the transient potassium current ($I_A$) and sustained potassium current ($I_K$) in cultured trigeminal ganglion neurons from TrpV1 KO mice. All together, these studies indicate that Cp can regulate the excitability of neurons independently of TrpV1 activation and, thus, through a mechanism involving the regulation of voltage-gated ion channels kinetics, Cp could induce the reduction in gamma power found in the present study.

## Activation state-selective effects on TrpV1-dependent vs TrpV1-independent mechanisms

Finally, similar to our findings, *Benninger et al. (2008)* have reported that the same concentration of Cp differentially regulates hippocampal activity depending on the activation state of the neural network. In the present study we found that 10 μM Cp can prevent Aβ-induced gamma impairment when hippocampal slices were treated before gamma oscillation induction (inactivated state). In contrast, the same Cp concentration failed to display protection when applied after gamma oscillations induction (activated state). In the study by *Benninger et al. (2008)*, the same Cp concentration we used (10 μM) induced a reduction in amplitude of evoked EPSC in dentate gyrus (activated state), but showed no effect on spontaneous EPSC (inactivated state) amplitude while significantly increasing their frequency. These results show how Cp can trigger different effects depending on the activation state of the network. This, together with our finding, suggests that Cp activation of TrpV1-dependent or TrpV1-independent mechanisms is influenced by the activation state of the network. Similarly, this dual effect of Cp has also been seen in trigeminal ganglion neurons in which Cp inhibition of VGSC and VGPC function has been shown to be mediated by TrpV1-dependent and TrpV1-independent mechanisms (*Cao et al., 2007*; *Yang et al., 2014*).

In conclusion, our study demonstrates that Cp treatment prevents the Aβ-induced impairment of hippocampal gamma oscillations through activation of the TrpV1 receptor. The cellular mechanism underlying the protective properties of TrpV1-activation involves the prevention of Aβ-induced

alterations of excitatory synaptic input and desynchronization of AP firing of CA3 PC. Even more important from a potential treatment perspective is our finding that TrpV1 receptor activation by Cp can also rescue functional network dynamics in hippocampus by reversing the Aβ-induced reduction of gamma oscillations power back to physiological control levels. This restoration of cognition-relevant network function holds the promise of potential cognitive benefit, which is the focus of ongoing and future studies. Our data are the first report of a preventive and restorative effect of TrpV1 receptor activation against Aβ-induced cytotoxicity in the hippocampal neuronal network and strongly suggest that TrpV1 activation is a suitable therapeutic target in the fight against AD.

# Materials and methods

## Key resources table

| Reagent type (species) or resource | Designation | Source or reference | Identifiers | Additional information |
|---|---|---|---|---|
| Genetic reagent (*Mus musculus*) | C57BL/6 | Charles River Laboratories | Strain Code: 027 | |
| Genetic reagent (*M. musculus*) | TrpV1 KO | Jackson Laboratory | Stock No: 003770 RRID:IMSR_JAX:003770 | PubMed: 10764638 |
| Peptide, recombinant protein | Recombinant Aβ1–42 | PMID: 25143621 | | Prof. Jan Johansson (Karolinska Institutet, Sweden) |
| Chemical compound, drug | Kainate Acid | Tocris Bioscience | Cat. No. 0222 | 100 nM |
| Chemical compound, drug | (*E*)-Capsaicin | Tocris Bioscience | Cat. No. 0462 | 5, 10 μM |
| Chemical compound, drug | Capsazepine | Tocris Bioscience | Cat. No. 0464 | 10 μM |
| Chemical compound, drug | Win55,212–2 mesylate | Tocris Bioscience | Cat. No. 1038 | 200 nM, 1 μM |
| Chemical compound, drug | JWH 133 | Tocris Bioscience | Cat. No. 1343 | 100 nM |
| Chemical compound, drug | Arachidonyl-2'-chloroethylamide | Tocris Bioscience | Cat. No. 1319 | 200 nM |
| Software | Spike phase-coupling code | Custom software | See Source code file 1 provided with this study | MATLAB script. Dr. André Fisahn (Karolinska Institutet, Sweden) |

## Drugs and chemicals

All chemical compounds used in intracellular and extracellular solutions were obtained from Sigma-Aldrich Sweden AB (Stockholm, Sweden). Receptor agonists, antagonists and other pharmacological compounds were obtained from Tocris Bioscience (Bristol, UK). (*E*)-Capsaicin (Cp), Capsazepine (Cz), Win55,212–2 mesylate (WIN), JWH 133 (JWH) and picrotoxin were dissolved in DMSO 100%. Arachidonyl-2'-chloroethylamide (ACEA) was dissolved in ethanol. Kainate acid (KA) was dissolved in milliQ water.

Recombinant Aβ1–42 was used in this study. Expression and purification of Aβ was previously reported (*Kurudenkandy et al., 2014*). Briefly, Met0Aβ1–42 was expressed in BL21*(DE3) pLysS Escherichia coli from synthetic genes and purified in batch format using ion exchange and passed through a 30,000 Da Vivas-pin concentrator filter (Sartorius Stedim Biotech GmbH) to remove large aggregates. Purified peptide was concentrated to 50–100 μM, aliquoted in low-bind Eppendorf tubes (Axygene) and stored at −20℃ until use. Before use Aβ was thawed on ice and briefly sonicated 5 min before application.

## Animals

Experiments were performed in accordance with the ethical permit granted by Norra Stockholms Djurförsöksetiska Nämnd to AF (N45/13). Animals used in this study included p17-30 C57BL/6 (WT) and TrpV1 knockout (TrpV1 KO) male mice (Charles River Laboratories and Jackson Laboratory, respectively). Animals were deeply anesthetized using isoflurane before being sacrificed by decapitation.

## Hippocampal slice preparation

Hippocampal slices were prepared as previously described (*Kurudenkandy et al., 2014*). Briefly, the brain was dissected out and placed in ice-cold artificial cerebrospinal fluid (ACSF) modified for dissection containing (in mM) 80 NaCl, 24 NaHCO3, 25 glucose, 1.25 NaH2PO4, one ascorbic acid, 3 Na pyruvate, 2.5 KCl, 4 MgCl2, 0.5 CaCl2, 75 sucrose and bubbled with carbogen (95% O2% and 5% CO2). Horizontal sections (350 µm thick) of the ventral hippocampi of both hemispheres were prepared with a Leica VT1200S vibratome (Leica Microsystems). After cutting, slices were transferred into a humidified interface holding chamber containing standard ACSF (in mM): 124 NaCl, 30 NaHCO3, 10 glucose, 1.25 NaH2PO4, 3.5 KCl, 1.5 MgCl2, 1.5 CaCl2, continuously supplied with humidified carbogen. The chamber was held at 37°C in a water bath during slicing and subsequently allowed to cool down to room temperature for at least 1 hr before commencement of experiments.

## Electrophysiology

For local field potential recordings (LFP), glass microelectrodes (4–6 MΩ) filled with standard ACSF were placed in CA3 stratum pyramidale. Interface chamber LFP were performed with a four-channel M102 amplifier (University of Cologne, Germany). Submerged chamber LFP and concomitant patch clamp recordings were performed using a Multiclamp 700B and Axopatch 200B amplifiers (Molecular Devices). Patch clamp (whole-cell) recordings were performed with borosilicate glass microelectrodes (4–6 MΩ) from visually identified CA3 PC using IR-DIC microscopy (Zeiss Axioskop, Germany and BX50WI Olympus, Japan). For AP firing and EPSC recordings ($V_h = -70$ mV), a potassium-based intracellular solution was used (in mM): 122.5 K-gluconate, 17.5 KCl, 4 Na$_2$ATP, 0.4 NaGTP, 10 HEPES, 0.2 EGTA, 2 MgCl, set to pH 7.2–7.3 with KOH. For IPSC recordings at $V_h = 0$ mV a cesium-based solution was used (in mM):. 140 CsMeSO4, 10 HEPES, 2 MgCl$_2$, 0.6 EGTA, 4 Na$_2$ATP, 0.3 NaGTP, set to pH 7.2–7.3 with CsOH.For IPSC recordings at $V_h$=-70 mV a high-chloride based solution was used (in mM):135 CsCl, 10 HEPES, 0.2 EGTA, 4 MgCl$_2$, 4 Na$_2$ATP, 0.3 NaGTP, 5 QX-314, set to pH 7.2–7.4.with CsOH; 50 µM picrotoxin was applied at the end of these experiments. APs recorded as single units (submerged chamber) were done using standard ACSF in the glass microelectrodes. Data were recorded with MultiClamp 700B and Axopatch 200B amplifiers, sampled at 10 kHz, conditioned using a HumBug 50 Hz noise eliminator (LFP signals only; Quest Scientific), low-pass filtered at 2 kHz, digitized (Digidata 1440A, Molecular Devices, CA) and stored on a hard disc using pCLAMP 10.0 software (Molecular Devices). In all experiments, gamma oscillations were elicited by applying kainic acid (KA, 100 nM) to the extracellular bath (*Fisahn et al., 2004*). Oscillations were allowed to stabilize for at least 20 min before recording.

## Data analysis

Fast Fourier Transformations for power spectra were calculated from 60s-long LFP data traces (segments of 8192 points) using Clampfit 10.2 software (Molecular Devices). Gamma power was then calculated as the integrated power spectrum between 20 and 80 Hz. Auto-correlograms were calculated in Clampfit using a 100 ms lag from the same LFP trace used for gamma power calculation. All LFP traces were pre-processed by applying a bandpass filter set to 15–60 Hz (high-pass: RC-single pole, low-pass: Gaussian) using Clampfit software. Auto-correlograms are presented as the average value of all the experiments.

Coefficient of rhythmicity (Cr) was calculated as a measure of the quality of gamma oscillations (*Andersson et al., 2010*; *Cangiano and Grillner, 2003*) and was defined as Cr=(α-β)/(α+β); α corresponding to the value of the height of the second peak and β to the first trough in the normalized auto-correlogram. Cr ranges between 0 and 1 in which the higher the coefficient, the more rhythmic the analyzed activity. Only recordings having a Cr ≥0.01 were considered rhythmic. In all cases, the

frequency predicted by the auto-correlogram was confirmed by the power spectrum of each recording segment analyzed.

EPSCs and IPSCs were detected off-line using MiniAnalysis software (Synaptosoft, Decatur, GA). Charge transfer, event amplitude and inter-event-interval (IEI) were analyzed using Excel software (Microsoft Office) and GraphPad Prism (GraphPad Software) with the results representing average values taken over 1 min period.

Spike phase-coupling analysis was performed on concomitant LFP and single-cell recordings using Matlab custom-written routines (see *Source code 1*) in order to relate the PC spiking activity to ongoing gamma oscillations (*Andersson et al., 2012*). To do this, 5-min-long segments from control and experimental condition recordings were used to perform the analysis. LFP were pre-processed using a band-pass filter set to 20–60 Hz (high-pass: RC-single pole, low-pass: Gaussian) in Clampfit. AP were detected by setting an amplitude threshold and the instantaneous phase of gamma oscillation was calculated using a Hilbert transform in order to determine the phase-angle at which each action potential occurred during ongoing oscillations. Phase-angles and gamma oscillation-phases were represented in polar-plots and expressed in radians with the peak of the oscillation cycle corresponding to $0\pi$ and the trough corresponding to $\pm\pi$ in the polar plots. In order to search for the synchronization level of AP firing, AP phase-angles frequency-distribution were normalized, a Gaussian function was fitted and the half-width at half-maximum (half-width) was then calculated as a measure of the synchronization level (*Kurudenkandy et al., 2014*). Considering that the more AP are fired on the same phase-angle the more synchronized the neuronal activity is, an increase in the half-width of the Gaussian curve fitted to the phase-angle distribution indicates a desynchronization of AP firing. The preferred phase-angle was calculated by averaging the AP phase-angles distribution of all the experiments and is represented by an arrow in the polar-plots. To test whether neurons fired in a phase-related manner, all concomitant recordings were tested for circular uniformity using Rayleigh's test. Only recordings with non-uniform circular distribution ($p < 0.05$) were considered for the analysis.

## Statistical analysis

All statistical analyses were performed using GraphPad Prism. Results are reported as mean ±SEM. Time-course data are presented normalized for comparison purposes between the different experimental conditions, for this data were binned and analyzed over 1 min period. and then normalized to the average of the 5 min of control recording before treatment application (for experimental data see source data files). Tests for statistical significance were performed on absolute values in all the experiments using Wilcoxon matched-pairs test (one-tailed) for paired data and Mann–Whitney U test for unpaired data (one-tailed). Significance was set at $p = 0.05$ for all statistical analyses. Data is presented as mean ±SEM. * indicates $p < 0.05$, **$p < 0.01$, ***$p < 0.001$ and n.s. indicates no statistically significant differences.

## Acknowledgements

This work was supported by the Swedish Research Council, the Swedish Brain Foundation, the Alzheimer Foundation, the Åhlén Foundation (André Fisahn), a KID PhD studentship grant (Daniela Papadia) and a CONACYT postdoctoral fellowship (Hugo Balleza-Tapia). The authors declare no competing financial interests.

## Additional information

### Funding

| Funder | Author |
| --- | --- |
| Vetenskapsrådet | André Fisahn |
| Alzheimerfonden | Jan Johansson |

The funders had no role in study design, data collection and interpretation, or the decision to submit the work for publication.

## Author contributions
Hugo Balleza-Tapia, Data curation, Formal analysis, Validation, Investigation, Methodology, Writing—original draft, Writing—review and editing; Sophie Crux, Yuniesky Andrade-Talavera, Daniela Papadia, Data curation, Formal analysis, Investigation; Pablo Dolz-Gaiton, Formal analysis; Gefei Chen, Jan Johansson, Resources; André Fisahn, Conceptualization, Resources, Supervision, Funding acquisition, Validation, Methodology, Project administration, Writing—review and editing

## Author ORCIDs
Hugo Balleza-Tapia (iD) http://orcid.org/0000-0002-3045-4594
André Fisahn (iD) http://orcid.org/0000-0003-1480-175X

## Ethics
Animal experimentation: Experiments were performed in accordance with the ethical permit granted by Norra Stockholms Djurförsöksetiska Nämnd to AF (N45/13). Animals used in this study included p17-30 C57BL/6 (WT) and TrpV1 knockout (TrpV1 KO) male mice (Charles River Laboratories and Jackson Laboratory, respectively). Animals were deeply anesthetized using isoflurane before being sacrificed by decapitation.

## Decision letter and Author response
Decision letter https://doi.org/10.7554/eLife.37703.024
Author response https://doi.org/10.7554/eLife.37703.025

# Additional files

### Supplementary files
• Source code 1. Spike phase-coupling code.
DOI: https://doi.org/10.7554/eLife.37703.020

• Source data 1. Unitary AP recordings.
DOI: https://doi.org/10.7554/eLife.37703.021

• Transparent reporting form
DOI: https://doi.org/10.7554/eLife.37703.022

### Data availability
Source data files have been provided for all figures, figure supplements, and the unitary action potential recordings.

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
