## [Decision Letter]

Thank you for submitting your article entitled "TrpV1 receptor activation rescues hippocampal neuron function and network gamma oscillations from Aβ-induced impairment" for consideration by *eLife*. Your article has been reviewed by two experts, and the evaluation has been overseen by a Reviewing Editor and a Senior Editor. Li-Huei Tsai, one of the two Reviewers, has agreed to reveal her identity.

The Reviewers have discussed their reviews with one another and the Reviewing Editor has drafted this decision to help you prepare a revised submission.

Summary:

An impairment in gamma oscillations accompanies the cognitive decline that is typical of Alzheimer's disease. The authors used a hippocampal slice system to investigate mechanisms underlying the β-amyloid-mediated impairments of hippocampal function and gamma oscillations. They report that activation of the TrpV1 receptor by capsaicin rescued those impairments. This study is of potential interest, even though its direct relevance for Alzheimer's disease is not clear. However, a number of points must be addressed satisfactorily, before this manuscript can be considered further.

Major points:

1) In Figure 1A, B, how was the dose (up to 1 μM) of the CB1/CB2 receptor agonist WIN determined? Could the lack of effect have been due to the low concentration of the drug? Was there any independent evidence showing activation of the receptors? Capsaicin, which did have an effect, was used at higher levels, up to 10 μM. What happened when WIN was used at 10 μM?

2) The effects of inhibiting TrpV1 on gamma power and rhythmicity should also be studied by including Aβ+Cz in Figure 1C-G.

3) The right-hand part of Figure 3E seems to show half-widths that are very similar between groups, except for one or two data points. Significance appears to rely on a single point. Can the authors please explain?

4) Figure 4 is confusing, in that capsaicin, which has been used at 10 μM before, is now used at 5 μM. Why this change in concentration compared to the previous experiments? Why do the authors say that they wanted to avoid an increase in concentration? Moreover, the message that Aβ causes desynchronisation of AP firing and that capsaicin can prevent this would be better supported by performing spike raster plots with PSTH/LFP, because this would make it possible to demonstrate more clearly the shift in synchronization.

5) Why was a different concentration of Aβ used in Figure 6? Please explain. In Figure 6C, the authors claim that TrpV1 activation rescued Aβ-induced degradation of gamma power. To better examine this, they may consider treating hippocampal slices with Aβ alone first, before adding capsaicin, to see if the latter can restore power back to wild-type levels.

---

## [Author Response]

Major points:1) In Figure 1A, B, how was the dose (up to 1 μM) of the CB1/CB2 receptor agonist WIN determined? Could the lack of effect have been due to the low concentration of the drug? Was there any independent evidence showing activation of the receptors? Capsaicin, which did have an effect, was used at higher levels, up to 10 μM. What happened when WIN was used at 10 μM?

We decided to test WIN at a low concentration (200 nM) because it has been reported that micromolar WIN concentrations (1, 10 µM) can reduce the power of gamma oscillations in the hippocampal area CA3 (Holderith et al., 2011), which would render our assay useless. In order to avoid such an effect on gamma oscillations we decided to use 200 nM WIN to test cannabinoid receptor activation since an even lower concentration (100 nM) had been reported to have cannabinoid receptor-dependent effects in hippocampal neurons (Straiker and Mackie, 2005).

Even though 1 µM WIN can impair gamma oscillations, we decided to test that concentration to investigate whether the lack of effect of WIN was due to the low concentration used (200 nM). As described in the Results section, this higher WIN concentration also failed to prevent the Aβ-induced degradation of gamma oscillations in our study.

We decided not to test even higher WIN concentration (for instance 10 µM) to avoid the impairing effect on gamma oscillations. Nonetheless, in order to corroborate our results gained using these low WIN concentrations, we tested two additional, sub-type-specific, cannabinoid receptor agonists: ACEA and JWH 133, which are CB1- and CB2-specific, respectively. As shown in Figure 1, by using these two specific CB1 and CB2 receptor agonists we found similar results compared to the CB1/CB2 agonist WIN and therefore concluded that cannabinoid receptor activation did not have any protective effect against the Aβ-induced degradation of gamma oscillations.

2) The effects of inhibiting TrpV1 on gamma power and rhythmicity should also be studied by including Aβ+Cz in Figure 1C-G.

We performed the requested experiments and have included the data description in the Results subsections “TrpV1 receptor activation prevents Aβ-induced impairment of functional network dynamics” and “TrpV1 receptor activation prevents Aβ-induced reduction of gamma oscillation rhythmicity”. Figure 1 as well as its figure legend were updated with the new Aβ+Cz data.

3) The right-hand part of Figure 3E seems to show half-widths that are very similar between groups, except for one or two data points. Significance appears to rely on a single point. Can the authors please explain?

Half-widths looked very similar between control and Aβ condition due to the high value in the Aβ group. As can be seen in the summary data sheet (Figure 3, source data files) in 5 out of 7 experiments Aβ increased the half-width.

To improve the ease of interpretation of this data set we have increased the n by adding 3 more experiments and excluded the experiment with the highest Aβ point value based on it now being a statistical out layer point according to the statistical analysis performed using Graph Pad Prism software. The figures and manuscript were updated in accordance with the new data: see textual additions in the subsections “Desynchronization of pyramidal cell action potentials and shift of excitatory/inhibitory current balance are responsible for impairment of functional network dynamics by Aβ” and “TrpV1 receptor activation prevents Aβ-induced desynchronization of pyramidal cell action potentials”, and new Figure 3 and legend. Please note that we reorganized Figures 3 and 4 following the additional experiments/analyses we performed in response to reviewers’ comment 4.

4) Figure 4 is confusing, in that capsaicin, which has been used at 10 μM before, is now used at 5 μM. Why this change in concentration compared to the previous experiments? Why do the authors say that they wanted to avoid an increase in concentration?

We investigated whether TrpV1 receptor activation by Cp was able to prevent Aβ impairment of the cellular mechanisms responsible for the generation of gamma oscillations by testing first 10 µM Cp in the submerged recording chamber. In contrast with the experiments done using the interface recording chamber (Figure 1 and 2), we found that 10 µM Cp was not able to prevent the Aβ-induced degradation of gamma oscillations (Figure 4—figure supplement 1).

As we mention in the Results section (subsection “TrpV1 receptor activation prevents Aβ-induced desynchronization of pyramidal cell action potentials”, second paragraph) even though the next approach would be to increase Cp concentration the literature indicates that higher Cp concentrations cause TrpV1-independent effects (Benninger et al., 2008; Boudaka et al., 2007; Cao et al., 2007; Lundbaek et al., 2005; Yang et al., 2014). We have provided an explanation and the references in the Results section (see the aforementioned subsection) and further strong arguments have been assessed in the Discussion section (subsection “Activation state-selective effects on TrpV1-dependent vs. TrpV1-independent mechanisms”).

Moreover, the message that Aβ causes desynchronisation of AP firing and that capsaicin can prevent this would be better supported by performing spike raster plots with PSTH/LFP, because this would make it possible to demonstrate more clearly the shift in synchronization.

In order to address the reviewer´s comment we have included new analysis in Figure 3B and Figure 4B and F: A filtered LFP recording showing three cycles of gamma oscillations and the frequency distribution plot of the AP gamma-phase-angles below the LFP signal. The frequency distribution plot shows the AP gamma-phase-angles repeated three times to visualize the AP firing windows during an ongoing gamma oscillation.

We believe that this display shows more clearly the synchronization level of the PC firing during ongoing gamma oscillations in control and experimental conditions. We also attempted to construct actual raster plots but found the visual results inferior to the one we chose above: Our cells fire too frequently for a visually informative raster plot to be constructed and the data cannot be pooled because every hippocampal slice preparation has a gamma oscillation of different frequency distribution. Tracking phase angles as chosen above appeared better because it is independent of slice gamma frequency/frequency variance. For changes in the manuscript see the subsection “TrpV1 receptor activation prevents Aβ-induced desynchronization of pyramidal cell action potentials” and new Figure 4 and legend.

5) Why was a different concentration of Aβ used in Figure 6? Please explain.

Aβ was used at 50 nM for experiments utilizing the interface recording chamber (Figures 1, 2 and 6 and Figure 1—figure supplement 1). When Aβ was applied acutely to the bath in experiments requiring the submerged recording chamber (Figures 3, 4 and 5, Figure 4—figure supplement 1 and Figure 5—figure supplement 1) its concentration was increased to 1 µM in order to offset the method-dependent lower signal amplitude in submerged conditions. This approach was discussed and vetted in one of our previous publications (Kurudenkandy et al., 2014).

Briefly, the signal-to-noise ratio in a setting where the tissue slice is completely submerged in ACSF (typical submerged recording chamber of patch-clamp set-ups) is much lower compared to a setting where the tissue slice sits at the interface between ACSF and humidified air (typical interface recording chamber). In the interface condition the tissue is electrotonically more compact (denser; higher gravity exerted on tissue suspended in air compared to liquid) and the hippocampal network delivers a bigger LFP signal. In submerged conditions the tissue is less compact and the greater volume of surrounding ACSF can shunt part of the signal leading to overall lower signal-to-noise ratio.

In Figure 6C, the authors claim that TrpV1 activation rescued Aβ-induced degradation of gamma power. To better examine this, they may consider treating hippocampal slices with Aβ alone first, before adding capsaicin, to see if the latter can restore power back to wild-type levels.

The experimental approach we used to study the Cp rescue of Aβ-induced degradation of gamma power was performed exactly as the reviewer suggests. To make this clearer we slightly altered the writing in the manuscript to differentiate the various experimental groups we used for Figure 6. We have re-written the Results section relating to Figure 6 to better explain the experimental conditions used (subsection “TrpV1 receptor activation rescues Aβ-induced impairment of functional network dynamics”).